



# ChinaCropSM1km: a fine 1km daily Soil Moisture dataset for Crop drylands across China during 1993–2018

Fei Cheng[1], Zhao Zhang[1, 2], Huimin Zhuang[1], Jichong Han[1], Yuchuan Luo[1], Juan Cao[1], Liangliang Zhang[1], Jing Zhang[1], Jialu Xu[1] and Fulu Tao[2, 3, 4]

[1]Academy of Disaster Reduction and Emergency Management Minsitry of Emergency Management & Ministry of Education, School of National Safety and Emergency Management, Beijing Normal University, Beijing 100875, China
[2]Key Laboratory of Land Surface Pattern and Simulation, Institute of Geographical Sciences and Natural Resources Research, Chinese Academy of Sciences, Beijing 100101, China
[3]College of Resources and Environment, University of Chinese Academy of Sciences, Beijing 100049, China
[4]Natural Resources Institute Finland (Luke), FI-00790 Helsinki, Finland

*Correspondence to*: Zhao Zhang (sunny_zhang@bnu.edu.cn)

**Abstract.** Soil moisture (SM) is a key variable of regional hydrological cycle and has important applications for water resource and agricultural drought management. Various global soil moisture products have been mostly retrieved from microwave remote sensing data. However, there is currently rare spatially explicit and time-continuous soil moisture information with a high resolution at a nation scale. Here we generated a 1km soil moisture dataset for stable crop drylands in China (ChinaCropSM1km) over 1993−2018 through random forest (RF) algorithm, based on numerous in situ daily observations of soil moisture. We used independently in situ observations (181327 samples) from the Agricultural Meteorological Stations (AMS) across China for training (164202 samples) and others for testing (17125 samples). An irrigation module was firstly developed according to crop type (i.e. wheat, maize), soil depth (0–10 cm, 10–20 cm) and phenology. We produced four daily datasets separately by crop type and soil depth, and their accuracy is all satisfactory (wheat $r$ 0.93, ubRMSE 0.033 $m^3m^{-3}$; maize $r$ 0.93, ubRMSE 0.035 $m^3m^{-3}$). The spatio-temporal resolutions and accuracy of ChinaCropSM1km are significantly better than those of global soil moisture products (e.g. $r$ increased by 116 %, ubRMSE decreased by 64 %), including the global remote-sensing-based surface soil moisture dataset (RSSSM) and the European Space Agency (ESA) Climate Change Initiative (CCI) SM. The approach developed in our study could be applied into other regions and crops in the world, and our improved datasets are very valuable for many studies and field managements such as agriculture drought monitoring and crop yield forecasting. The data are published in Zenodo at https://zenodo.org/record/6834530 (wheat$_{0–10}$) (Cheng et al., 2022a), https://zenodo.org/record/6822591 (wheat$_{10–20}$) (Cheng et al., 2022b), https://zenodo.org/record/6822581 (maize$_{0–10}$) (Cheng et al., 2022c) and https://zenodo.org/record/6820166 (mazie$_{10–20}$) (Cheng et al., 2022d).

## 1 Introduction

Soil moisture (SM) is closely associated with droughts and floods, consequently agricultural productions (Tao et al., 2003). Thus, SM information at a high resolution is critical to improve crop yield prediction (Prasad et al., 2006; Chakrabarti et al.,



2014) and drought impact assessment (Sheffield, 2004). However, such higher resolution in both temporal (e.g. daily and more than decade) and spatial scales are still unavailable across China, especially for dry croplands.

SM can be obtained by several ways, including in situ observations (Walker et al., 2004; Bogena et al., 2007), remote sensing retrieval (Mohanty et al., 2017; Wei et al., 2019), and process-based model simulations (Vergopolan et al., 2020; Ahmed et al., 2021). The field observations provide the most accurate SM but being expensive and time-consuming, and large uncertainties from extrapolating the limited observations into larger regions with high heterogeneity (Collow et al., 2012; Crow et al., 2012). The microwave sensors have been applied to retrieve SM in recent years (Schmugge et al., 2002; Wigneron et al., 2003; Amazirh et al., 2018). The microwave sensors can only monitor near-surface SM (0−10 cm) (Eagleman and Lin,

1976; Jackson et al., 1982). The passive microwave sensors can monitor daily SM but with a coarse resolution (25−40 km), comparing with a high spatial resolution (10–30 m) whereas a coarser repetition interval (15−25 days) for active ones (Eagleman and Lin, 1976; Jackson et al., 1982; Mallick et al., 2009). Such SM products have large uncertainties due to the limitations from satellite coverage and downscaling methods, although they can easily cover large regions compared with the in situ observations (Loew et al., 2013; Su et al., 2016; Peng et al., 2017). Deriving the SM from model simulation is also

challenging because of its high requirements in input data, computing ability and large uncertainties from model parameters (Wang and Qu, 2009; Yilmaz et al., 2012; Petropoulos et al., 2015). In addition, many studies have found that irrigation, as an additional water supply source other than precipitation, reduces soil albedo (Chen and Dirmeyer, 2019), increases heat capacity (Wang et al., 2019), alters local SM (Lawston et al., 2017), and affects the water/energy budget (Shen et al., 2013). However, few studies have taken irrigation into account in developing SM data products at the national or global scales (Drewniak et al.,

2013; Qiu et al., 2016a). Thus, it is critical yet challenging to improve SM accuracy at both spatial and temporal resolutions. As one part of the Climate Change Initiative (CCI), the European Space Agency (ESA) published a long-term surface SM dataset, and the latest version (v06.1) covered the period of 1978–2020 (https://www.esa-soilmoisture-cci.org/, last access: 10 Apr. 2022) (Dorigo et al., 2017a; Gruber et al., 2019; Preimesberger et al., 2021). The ESA CCI SM p*r*oducts are consistent with the observed values at some grassland and farmland sites in China (Liu et al., 2011; Albergel et al., 2013; Dorigo et al.,

2015, 2017b), however, they have a coarse spatial resolution (~27 km) and lots of coverage gaps (Llamas et al., 2020; Guevara et al., 2021). More recently, based on multiple neural networks, the global remote-sensing-based surface soil moisture (RSSSM) dataset covering 2003–2018 at 0.1° resolution was developed by using Soil Moisture Active Passive (SMAP) SM as the primary training target. RSSSM improved Coefficient of determination ($r^2$) of 0.46 and Root Mean Squared Error (RMSE) of 0.083 m³ m⁻³, with a 10-day resolution (Chen et al., 2021). In 2020, another new SM dataset in China from 2002 to 2018 was

provided from different passive microwave SM products and model-based downscaling techniques (Meng et al., 2021). With an improved correlation coefficient $r$ of 0.84 and an unbiased root-mean-squared error (ubRMSE) of 0.056 m³ m⁻³, the new dataset has a 0.05° spatial resolution and a monthly time resolution. These SM products did contribute largely to related agriculture studies and managements, however, they are still too coarse to assess agricultural drought risk and predict crop yield accurately.

Despite numerous efforts have been devoted to develop SM products, major concerns should be addressed: (1) agricultural management activities such as irrigation have not been fully considered by the previous studies, especially in countries like China with extensive areas irrigated (Zhu et al., 2013); (2) both spatial and temporal (e.g. daily) resolutions of SM products need to be improved for regional agricultural managements; (3) the SM accuracy need to be further improved. In recent years, the in situ observations are becoming available (Li et al., 2005). Some new methods such as machine learning are increasingly

applied to many fields and have been shown to be robust in incorporating multiple sources data to develop spatiotemporal datasets (Ahmad et al., 2010; Srivastava et al., 2013; Im et al., 2016).

Thus, our main objectives in the study are: to develop a novel method to generate a daily 1km SM dataset for dry croplands across China based on numerous field observations; to evaluate their accuracy and compare them with current products; to explore the spatio-temporal characteristics of soil moisture for crop drylands. We are sure our methods and datasets will be

valuable for agriculture drought monitoring and crop yield forecasting.

## 2 Materials and Method

### 2.1 Study area

The study area is dominated by dryland crops such as wheat and maize in China, with complex cultivation methods (Wu and Li, 2012) and various irrigation activities (Huang et al., 2015). According to the annual crop harvesting areas of crops across

mainland China from 2000 to 2015 (Luo et al., 2020a, b), maize and wheat are the two main crops in China, accounting for 35.4 % of the total harvested area (FAOSTAT, 2019). The study areas and SM in situ field monitoring sites for the two crops are shown in Figure 1.

### 2.2 Data

### 2.2.1 In situ SM observations

The in situ SM observation data (http://data.cma.gov.cn/data/detail/dataCode/AGME_AB2_CHN_TEN.html, last access: 18 April 2021) from 1993 to 2018 in this study were obtained from agriculture meteorological sites in China, which recorded the location, crop type, phenology, soil depth and SM. SM was measured at the depths of 10 cm and 20 cm at each agro-meteorological station on the 8th, 18th and 28th of each month. For each sample, crop phenology was observed and recorded by well-trained agricultural technicians in experimental fields (the average field size was 0.15 ha) and then were checked and

qualified by the Chinese Agricultural Meteorological Monitoring System (CAMMS). The first layer (0–10 cm) has been widely used to investigate spatial and temporal characteristics of SM and validate SM retrieved from microwave across China (Lacava et al., 2012; Zeng et al., 2015; Liu et al., 2018; Fang et al., 2020).



We collected the in situ observations of maize (287 sites) and wheat (240 sites), with total 181327 samples (maize: 36226 samples for the 0–10 cm soil layer, 36245 samples for the 10–20 cm soil layer; wheat: 54396 samples for the 0–10 cm soil layer, 54460 samples for the 10–20 cm soil layer).

### 2.2.2 Environmental factors

The environmental factors are classified into Site features and Gridded features, which both include meteorological data (MD), day of year (DOY), Classified Irrigation (CIR), soil properties (SP), remote sensing data (RSD), and geographical information (GI) (Table 1).

MD includes daily total precipitation (pre) and accumulated precipitation for 10 days (pre10) from the meteorological stations across China (CNMSs) (http://data.cma.cn, last access: 10 April 2021) (Figure S8).

CIR was calculated using Eq. (1).

$$\text{CIR} = \begin{cases} 1, C_i P_j D_k SM \geq SMI_{ijk}; \\ 0, C_i P_j D_k SM < SMI_{ijk}; \end{cases} \tag{1}$$

where $C_i$, $P_j$, $D_k$, and $SMI_{ijk}$ are crop type, phenology, soil depth, and the evaluation index of relative soil moisture (SMI) corresponding to the crop type $i$, phenology $j$, soil depth $k$. SMI is a threshold to determine when irrigation is applied (Table 2), which was released by Ministry of Water Resources of China (CNMWR) (http://www.mwr.gov.cn, last access: 10 July 2022) in July 2012.

SP includes sand, silt, gravel, organic carbon, clay contents, soil PH and bulk density, obtained from Harmonized World Soil Database Version 1.2 (http://webarchive.iiasa.ac.at/Research/LUC/External-World-soil-database/HTML/, last access: 18 Aug. 2021). The original 30 arc-second raster spatial resolution data were resampled to a 1 km resolution based on the nearest neighbor interpolation, and the site-related SP were extracted from values to points using ArcGIS 10.5 software (ESRI).

RSD includes Reference Evapotranspiration (pet) and field capacity (fc). pet was obtained from TerraClimate (https://doi.org/10.7923/G43J3B0R, last access: 28 Aug. 2021) which included monthly climate and climatic water balance from 1958-present with a resolution of 1/24° or ~4 km (Abatzoglou et al., 2018). fc was obtained from OpenLandMap (https://zenodo.org/record/2784001, last access: 18 July 2021) (Hengl and Gupta, 2019) which included fc under 33kPa at 0 cm (b0) and 10 cm (b10) depth.

GI includes latitude (lat), longitude (lon), moisture index (im) (Thornthwaite, 1948), and river vector data, provided by the Data Centre for Resources and Environmental Sciences, Chinese Academy of Sciences (http://www.resdc.cn/Default.aspx, accessed on 18 April 2021). The distance from each AMS to river networks at all levels (R4, R5, R12) in China was calculated using Euclidean distance analysis method.

### 2.2.3 Public SM products for comparison

We used two existing SM products for comparison: 1) The ESA CCI SM data are a merged multi−satellite surface SM product, which consists of active, passive, or combined products. The SM retrievals are from four microwave radiometers (SMMR:



Scanning Multichannel Microwave Radiometer, SSM/I: Special Sensor Microwave/Imager, TMI: Tropical Rainfall Measuring

Mission (TRMM)'s Microwave Imager, and AMSR-E: Advanced Microwave Scanning Radiometer for the Earth Observing

System) and two scatterometers (AMI: Active Microwave Instrument, ASCAT: Advanced Scatterometer) in a 0.25° global

daily dataset. The data assimilated relies on their respective sensitivity to vegetation density and uses a Global Land Data

Assimilation System (GLDAS) surface SM product (Rodell et al., 2004) as a climatology reference (Wagner et al., 2012). The

active/passive products are the integration of the scatterometer /radiometer-based SM retrievals, respectively, while ESA CCI

SM product is the fusion of both the active and passive products. We used the v05.2 product for comparison because of its

advantages comparing with active/passive products (Liu et al., 2012; Dorigo et al., 2017c). 2) The RSSSM is an improved

global    long-term    remote-sensing-based    surface    SM    dataset    covering    2003–2018    at    0.1°    resolution

(https://doi.org/10.1594/PANGAEA.912597). Considering their compatibility, we chose 1995 to 2018 for comparison between

ChinaCropSM1km and ESA CCI SM, and the 2003–2018 period for that of ChinaCropSM1km and RSSSM.

**2.3 Method**

**2.3.1 Variable selection and data treatment**

For the site-related variables, we deleted those with high multicollinearity ($|r| > 0.5$) according to the factor stacks (Figure S2

and S3). Thus, the 11 independent variables (pre, pre10, DOY, CIR, T_REF_BULK, R4, im, pet, lat, lon and fc) were selected

because they well characterize the impacts of meteorological, temporal, irrigation, soil properties, geographical information

on regional SM. We used the "Euclidean distance" option of the Spatial Analyst Tools in ArcGIS10.5 to obtain the variables

related with river network in China (Danielsson, 1980). We also applied the kriging interpolation method to obtain

precipitation-related variables (e.g. pre and pre10) from CNMSs in China. Thereafter, all gridded maps were processed in the

WGS84 UTM zone 45N Geographic Coordinate System (EPSG:32645), and resampled to the same spatial resolution (1 km).

**2.3.2 Model development**

Ensemble learning was used to aggregate a collection of algorithms to predict the potential impacts, which represents a better

method than that uses any algorithm alone (Brownlee, 2016). Random Forest (RF) is a typical ensemble learning algorithm

which can be used to build predictive models for both classification and regression purposes. RF fits an ensemble of models

that first train a multitude of decision trees and then obtain predictions by an average or vote through all individual trees

(Breiman, 2001). The algorithm introduces extra randomness when growing trees and searches for the best trees among a

random subset of features. This technique results in greater tree diversity, generally yielding an overall better model (Hutengs

and Vohland, 2016; Lagomarsino et al., 2017). In addition, bagging method, which constructed multiple training sub-dataset

by resampling with replacement of the original dataset, is employed to reduce the variance and overfitting (Díaz-Uriarte and

Alvarez de Andrés, 2006; Zhang et al., 2018). Its high accuracy and stability in agricultural fields have been substantiated in





several previous studies, especially for predicting grain yield, identifying crop planting areas, and mapping soil properties

(Hengl et al., 2015; Jeong et al., 2016; Sun et al., 2019).

Hyper parameters in a RF model are very important to optimize its performance. Such parameters are initially defaulted, and we need investigate their appropriateness or find a potentially better values during developing a RF regression (RFR). The important hyper parameters include follows:

n_estimators: the number of trees that the algorithm builds before taking the maximum voting or average over predictions. A

high number of trees increases the performance and makes the predictions more stable, but demand more computations.

max_features: the maximum number of features that the random forest considers on a per-split level. The condition is based on variance for regression.

min_samples_leaf: the minimum number of leafs that are required to split an internal node.

max_samples: ratio of samples needed for training each tree.

We applied the 10-fold cross-validation method to tune the four hyper parameters for avoiding the overfitting of RF models (Figure S4). Meanwhile, we use this 10-fold cross-validation to evaluate model performance (Figure S7).

The detailed irrigation module is shown in Figure S1. Given SM is highly sensitive to irrigation application for crop drylands in China, we firstly used the RF classification (RFC) to build irrigation module. This module aims to predict whether irrigation application occurred there, and assign response variable "1" for irrigation and "0" for without-irrigation according to the

response variables and predictor ones (the same environmental indicators used in producing ChinaCropSM1km).

As for the response variable (Classified Irrigation CIR) is calculated by irrigation threshold (Table 2) and in situ information, including crop type, phenology and soil depth. Then we used the forecasted CIR as an additional predictor, integrating with other key predictor variables, to drive RFR for forecasting SM. Considering the regional differences in SM, we randomly sampled in situ SM observations (90% for training and 10% for testing) in each agricultural zone to develop RF model. Totally,

98576 (65626) and 10820 (6845) observations were used for training and testing the model for wheat (maize), respectively (Figure 2).

The hyper parameters in the optimal model were determined as 50, 1, 4 and 1 for the respective n_estimators, max_samples, min_samples_leaf and max_features according to the highest accuracy during training (Figure S4). We implemented this processes in Matlab9.8.0 (R2020a). More information could be referred to the MATLAB help center

(https://www.mathworks.com/help/stats/regressionlearner-app.html, last access: 26 May 2022).

The features importance was evaluated for the RF model with the greatest regression accuracy by ordering the out-of-bag predictor observations, using the Matlab '*oobPermutedPredictorImportance*' function (https://www.mathworks.com/help/stats/regressionbaggedensemble.oobpermutedpredictorimportance.html, last access: 26 May 2022). We also used the method to measure the importance of each predictor variable during predicting ChinaCropSM.




### 2.3.3 Evaluation metrics for validation and comparison

The in situ observations provide the most accurate SM, all performance measures were calculated using the testing dataset for evaluation purposes. All SM products were evaluated against the in situ observations (testing dataset) according to three metrics: Root Mean Square Error (RMSE; $m^3m^{-3}$), bias ($m^3m^{-3}$), unbiased RMSE (ubRMSE; $m^3m^{-3}$), Explained variance($R^2$), and the correlation coefficient ($r$), which are defined as follows Eqs. (2)–(6):

$$r = \frac{1}{N-1}\sum_{i=1}^{N}\left(\frac{P_i-\bar{P}}{\sigma_P}\right)\left(\frac{O_i-\bar{O}}{\sigma_O}\right) \tag{2}$$

$$R^2 = 1 - \frac{\sum_{i=1}^{N}(P_i-O_i)^2}{N\sum_{i=1}^{N}(P_i-\bar{P})} \tag{3}$$

$$bias = \frac{1}{N}\sum_{i=1}^{N}(P_i - O_i) \tag{4}$$

$$RMSE = \sqrt{\frac{1}{N}\sum_{i=1}^{N}(P_i - O_i)^2} \tag{5}$$

$$ubRMSE = \sqrt{RMSE^2 - bias^2} \tag{6}$$

where the overbar indicates the mean; $P_i$ is the i[th] prediction SM from products; $O_i$ is the i[th] in situ observation SM; $N$ is the total number of observations; and $\sigma_o$ and $\sigma_p$ are the standard deviations of the in situ observed or predicted SM, respectively. In addition, we compared our four subset data with RSSSM and ESA CCI SM separately through evaluating their spatial and temporal accuracy related to in situ surface SM observations (Table S2 and S3).

## 3 Results and discussion

### 3.1 ChinaCropSM1km products validation

The scatterplots between the predicted SM and those observations were displayed by soil layers and crops (Figure 3). We found that the SM predicted by RF model agreed well with in situ SM observations, with ubRMSE of 0.028–0.037, bias of -0.0011–0.0009, $r$ from 0.925–0.944. Moreover, the mean bias in predicting SM for wheat was negative (Figure 3-a, b), while those for maize were positive (Figure 3-c, d). These findings suggest that maize SM were overestimated while those for wheat were underestimated. The absolute value of mean bias and RMSE in predicting SM at top soil depth (0–10 cm) for both crops were relative larger (eg. RMSE 0.036>0.028) than that for soil depth of 10–20 cm. It indicates that RF model performed better in predicting the SM content at 10–20 cm layer than that at 0–10 cm layer, which was consistent with the previous studies (O and Orth, 2020).

### 3.2 The improvement of ChinaCropSM1km products with an irrigation module

Interestingly, all prediction accuracy of SM were consistently improved both for crops and depths (Figure 4) with comparison of those without an irrigation module (Figure 4). Specifically, $R^2$ values were increased by 6.8–9.7%, RMSE were decreased by 16–23% (Figure 4). Among these, $R^2$ values for maize SM were slightly improved than those of wheat and RMSE for maize





was decreased more than that of wheat. Such finding further suggests that the irrigation water requirements of maize are higher

than for wheat, which was consistent with the reason that summer maize requires large amounts of water to produce high

yields. (Mohammed Karrou, 2012).

### 3.3 The significant scores of different factors for simulating SM

It is critical to select which independent variables involved into a model, neither too many or too less, simultaneously avoiding

multicollinearity among them. We have deleted 7 variables due to their high correlations ($|r| > 0.5$) with the 11 variables selected

(Figure S2 and S3). Surprisingly, the top first was scored to irrigation factor (CIR), followed by pre10 (ante-accumulated

precipitation over ten days) and fc (field capacity) (Figure 5). Current daily precipitation show significantly different

importance on SM planted by wheat and maize, with the similarity for DOY. Nevertheless, all other factors show less

importance on SM simulation. Comparing with the significant roles of precipitation-related variables (e.g. pre10, pre) on SM

in most rainfall-fed areas, however, irrigation shows overwhelming impacts on dryland soil moisture across China (Qiu et al.,

2016b). Such result highlights more accurately monitoring management activities, including irrigating times, areas and

quantity, will further improve irrigation module, consequently improve SM simulation (Wu et al., 2020; Zhang et al., 2015,

2022).

### 3.4 The temporal and spatial patterns between ChinaCropSM1km and the in situ SM observations

The SM values in ChinaCropSM1km are significantly correlated with the in situ SM observations, with a mean $r$ of 0.92, 0.94,

0.93 and 0.94, respectively, for wheat$_{0-10}$, wheat$_{10-20}$, maize$_{0-10}$ and maize$_{10-20}$, during the whole growing period (Figure 6).

The spatial coefficients for wheat at 10−20 cm are generally higher than the surface SM (0.94 vs. 0.92), and the two soil depths

SM in April and September are significantly higher (Figure 6–a, b). We attributed the high spatial correlations of surface SM

to irrigation impacts, because April and September are planting time for both spring and winter wheat. The better relationships

further substantiated the irrigation module developed in our SM model improves the simulation accuracy for surface SM.

Consistently, the spatial coefficients for maize at 10–20cm depth are higher than those for 0−10cm (0.94 vs. 0.93) (Figure7-c,

d). At the sowing (Apr.), heading (Jul.), and milking (Aug.) stages, maize usually demanded water supplying a lot. The spatial

coefficient for maize SM at both soil depth from May to Aug. were lower than the mean value potentially due to the lack of

irrigation applications (Yin et al., 2016) (Figure 6).

We still further analyzed the temporal pattern of SM accuracy in different regions (Figure 7). The median of $r$ values for

Huang–Huai–Hai Plain and Northern Arid and Semiarid region were higher than that in other agricultural regions because of

larger training samples. Our findings further substantiated that a larger training sample size will cause a higher temporal

accuracy, indicated by a higher $r$ and a lower RMSE (Figure S6). However, the poor performance in Yunnan–Guizhou Plateau

might be caused by smaller training samples (Figure S6).



**3.5 Comparisons between ChinaCropSM1km and public global SM products**

We further compared our ChinaCropSM1km with the two popular products through evaluating their spatial-temporal accuracy related to in situ surface SM observations. We summarized their evaluation indexes by individual product in Table 3, which indicated consistently the bolds for our ChinaCropSM1km (all $r$ >0.90, RMSE <0.04), while RSSSM and ESA CCI SM were shown by $r$ <0.50 and RMSE >0.1.

To match the different spatial resolutions of three products, we calculated the averages of all in-site observations in the same

pixel (e.g. 1 km, 27 km or 0.1°) to make their spatial-temporal accuracy comparable. Interestingly, all indexes of our products were consistently indicated by the higher accuracy (e.g. $r$ 0.94, bias 0.005, RMSE 0.034, ubRMSE 0.034) (Figure 9). RSSSM dataset significantly underestimated SM with an averaged bias of −0.114, companying with a higher RMSE of 0.150. ESA CCI SM performed better than RSSSM (e.g. RMSE 0.11 vs. 0.15) derived from Soil Moisture Active Passive (SMAP) (Entekhabi et al., 2010), and we ascribed such improvement partly into some corrections based on in situ observations for ESA

CCI SM (Dorigo et al., 2017b). Such results highlight SM products derived solely from remote sensing satellites should be corrected with ground observations. Moreover, neither RSSSM nor ESA CCI SM had considered the irrigation activities, thus their spatial correlations with ground observes are incomparable to those of our products ($r$ 0.944 vs. 0.381 0.256) (Figure8). Our study substantiates strongly that an irrigation module should be taken into account when developing SM simulation models of producing SM products.

**4 Data availability**

The 1km gridded daily Soil Moisture for Croplands in China (ChinaCropSM1km) is available at https://zenodo.org/record/6834530 (wheat$_{0-10}$) (Cheng et al., 2022a), https://zenodo.org/record/6822591 (wheat$_{10-20}$) (Cheng et al., 2022b), https://zenodo.org/record/6822581 (maize$_{0-10}$) (Cheng et al., 2022c) and https://zenodo.org/record/6820166 (mazie$_{10-20}$) (Cheng et al., 2022d).

**5 Discussion and Conclusions**

We developed a daily 1km soil moisture dataset based on numerous field observations (181327 samples) from 1993–2018, which significantly enrich the current SM datasets available. Our ChinaCropSM1km shows higher temporal-spatial resolution and accuracy than the popular global SM products. Moreover, to date, few studies have provided a daily SM product with such higher resolution, combining different soil depths and an irrigation module. ChinaCropSM1km is the first SM product with a

higher spatial resolution (~1km) at 0–10, 10–20 cm depth in China croplands by compiling the ground observations and using RF method.

Our ChinaCropSM1km predicted by RF model agreed well with in situ SM observations (ubRMSE ranges from 0.028–0.037, bias ranges from −0.0011–0.0009, $r$ ranges from 0.925–0.944, and $R^2$ ranges from 0.860–0.895). An irrigation module was

firstly developed according to crop type (i.e. wheat, maize), soil depth (0–10 cm, 10–20 cm) and phenology. All prediction
accuracy of SM were consistently improved ($R^2$ values were increased by 6.8~9.7%, RMSE were decreased by 16~23%) both
for crops and depths. Meanwhile, ChinaCropSM1km generally has advantages over other popular gridded SM products
(RSSSM and ESA CCI SM) through evaluating their spatial-temporal accuracy related to in situ SM as the benchmark. Our
ChinaCropSM1km has relatively higher accuracy (all $r$ >0.90, RMSE <0.04), while RSSSM and ESA CCI SM were shown
by $r$ <0.50 and RMSE >0.1.

Our method for generating cropland SM is applicable to other regions and crops, but the environmental data will be
increasingly required considering the SM variabilities are complex processes controlled by many factors (Famiglietti et al.,
2008; Qin et al., 2013; Guevara and Vargas, 2019), especially for irrigation activities. For example, to characterize more
accurately the irrigation activities, more field samples are highly required in both spatial and temporal resolutions. Other
auxiliary data on information of crop growth, classification, and managements will benefit to develop an irrigation module and
derive SM datasets more accurately. Moreover, advanced algorithms will be potential alternative of RF because of its strong
dependence on the input data (Breiman, 2001; Rasmussen, 2004). Finally, we are sure more accurate SM dataset will be
produced by applying the approach into other crops and areas in future with all above improvements.

## Author contributions

FC, HZ, ZZ, JH, JC, YL, LZ, JZ and JX contributed to the design of this research; FC and ZZ collectively prepared the
manuscript with contributions from all co-authors; JH, JC, YL, LZ, JX and JZ revised the manuscript; FC and HZ developed
the model code.

## Competing interests

The authors declare that they have no conflict of interest.

## Financial support

This study was supported by the National Key Research and Development Program of China (grant no. 2020YFA0608201)
and the National Natural Science Foundation of China (grant no. 41977405).

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



**Figure 1 Study areas and the SM in situ field monitoring sites in China. NAS: Northern Arid and Semiarid region; LP: Loess Plateau; HP: Huang–Huai–Hai Plain; SCB: SiChuan Basin; MYP: Middle–lower Yangtze Plain; YGP: Yunnan–Guizhou Plateau and southern China; QT: Qinghai–Tibet region; ChinaCropland1km: the harvesting areas of crops across mainland China.**



**Table 1 Environmental factors used in the study, including meteorological data (MD), day of year (DOY), irrigation data (CIR), soil properties (SP), remote sensing data (RSD), and geographical information (GI).**

| Data type | Variable | Data description | Temporal resolution | Spatial resolution |
|---|---|---|---|---|
| MD | pre | daily precipitation | daily | 1 km |
| | pre10 | ante-accumulated precipitation over ten days | daily | 1 km |
| DOY | DOY | day of year | daily | 1 km |
| CIR | CIR | classified irrigation | - | - |
| SP | T_REF_BULK | unit: %kg dm$^{-3}$. | - | 1 km |
| | T_SAND | unit: % wt. | - | 1 km |
| | T_CLAY | unit: % wt. | - | 1 km |
| | T_PH_H2O | unit: %-log ($H^+$). | - | 1 km |
| | T_GRAVEL | unit: % vol. | - | 1 km |
| | T_SILT | unit: % wt. | - | 1 km |
| RSD | pet | potential evapotranspiration | monthly | 4 km |
| | fc | field capacity | - | 250 m |
| GI | R4 | river network vector I | - | - |
| | R5 | river network vector II | - | - |
| | R12 | river network vector III | - | - |
| | lat | latitude | - | - |
| | lon | longitude | - | - |
| | im | moisture index | - | - |

Note: REF_BULK: soil bulk density; PH_H2O: hydrogen ion concentration; GRAVEL: volume percentage of crushed stone; T: the topsoil layer. The dash line represents no default values.





**Table 2 Evaluation index of relative soil moisture (SMI) in different growth periods of crops at 0–10, 10–20 cm depth.**

| SMI in different growth periods of wheat (%) | | | | | | | |
|---|---|---|---|---|---|---|---|
| seeding | seedling | tillering | greening | jointing | booting | grouting | mature |
| 70~90 | 75~95 | | | 80~95 | | 55~60 | |
| SMI in different growth periods of maize (%) | | | | | | | |
| seeding | seedling | jointing | booting | tasseling | grouting | mature | |
| 75~85 | 65~75 | 70~80 | 75~85 | | 65~75 | | |





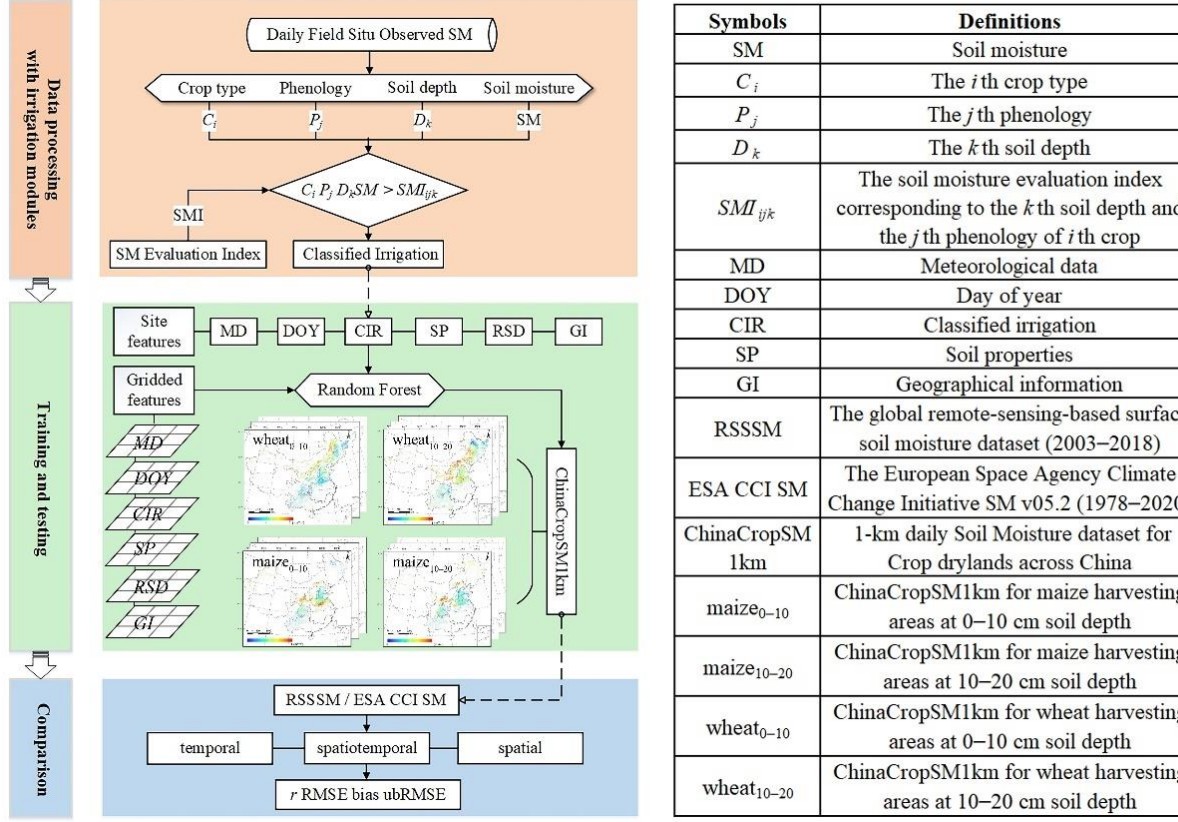

**Figure 2 Flow chart for producing ChinaCropSM1km with an irrigating module.**

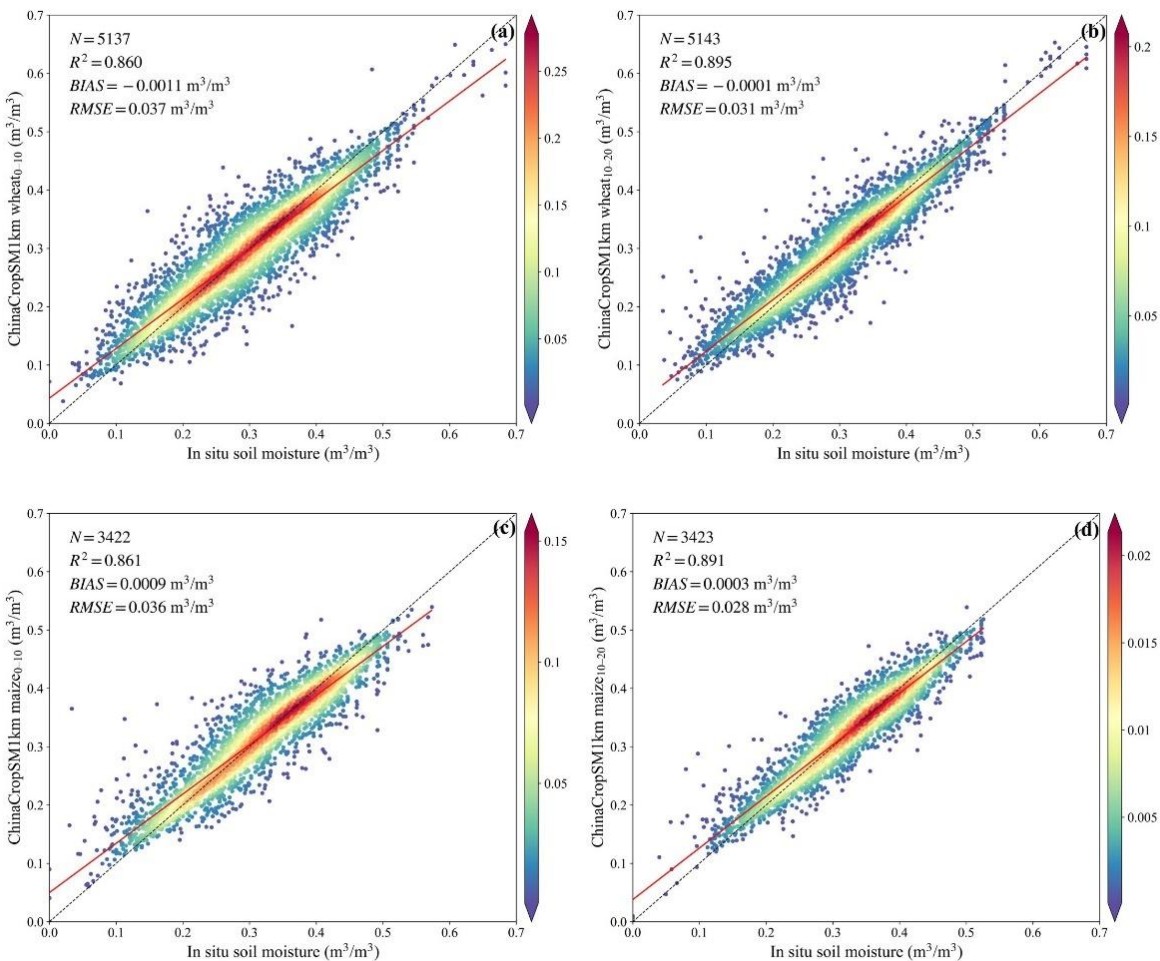

**Figure 3 Comparison between the predicted soil moisture (ChinaCropSM1km) and in situ samples by crops and depths (cm). (a) wheat$_{0-10}$, (b) wheat$_{10-20}$, (c) maize$_{0-10}$ and (d) maize$_{10-20}$. The red lines are the trend lines, the colorbar means point density, and the black lines for 1:1 lines.**





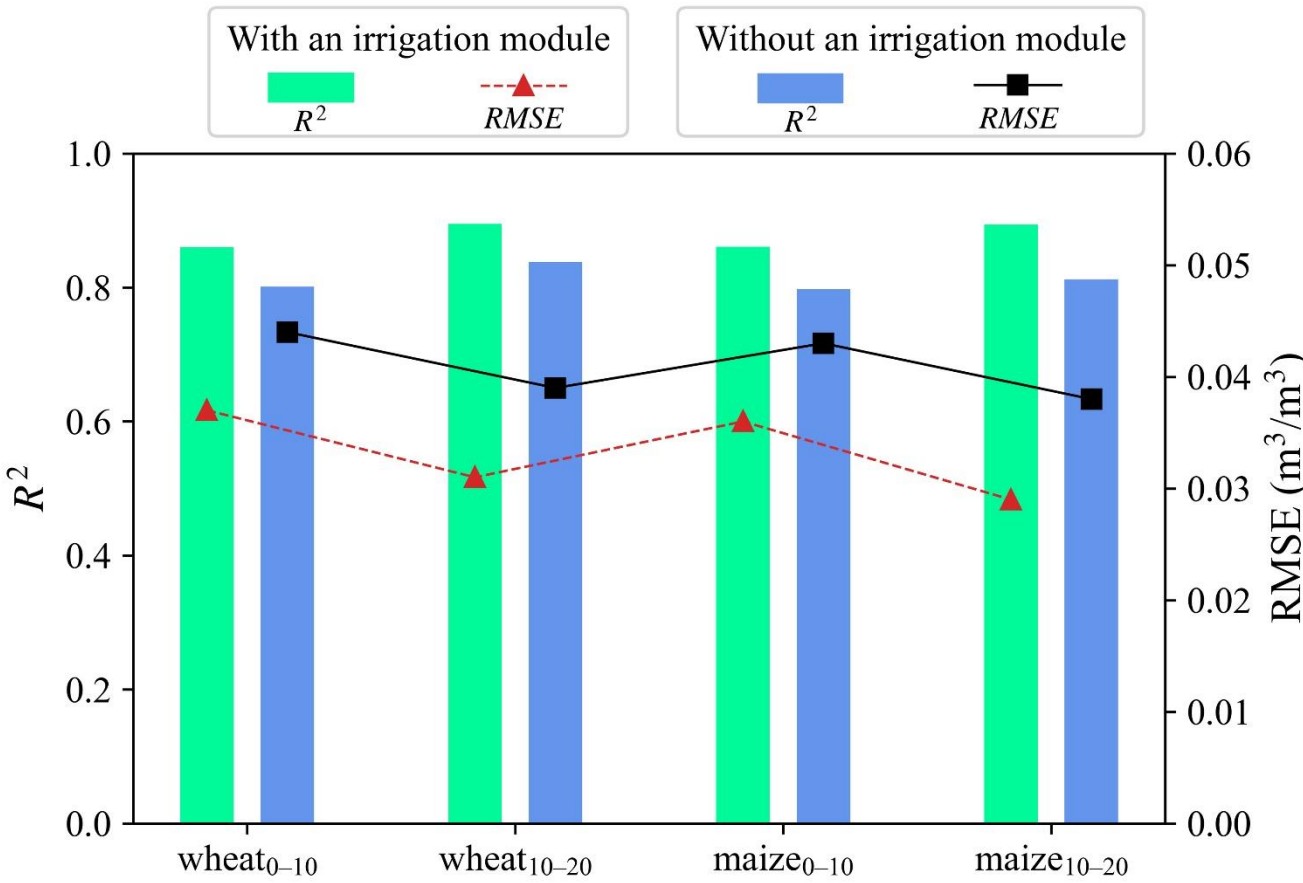

**Figure 4. Comparison of soil moisture accuracy between with irrigation and without an irrigation module.**





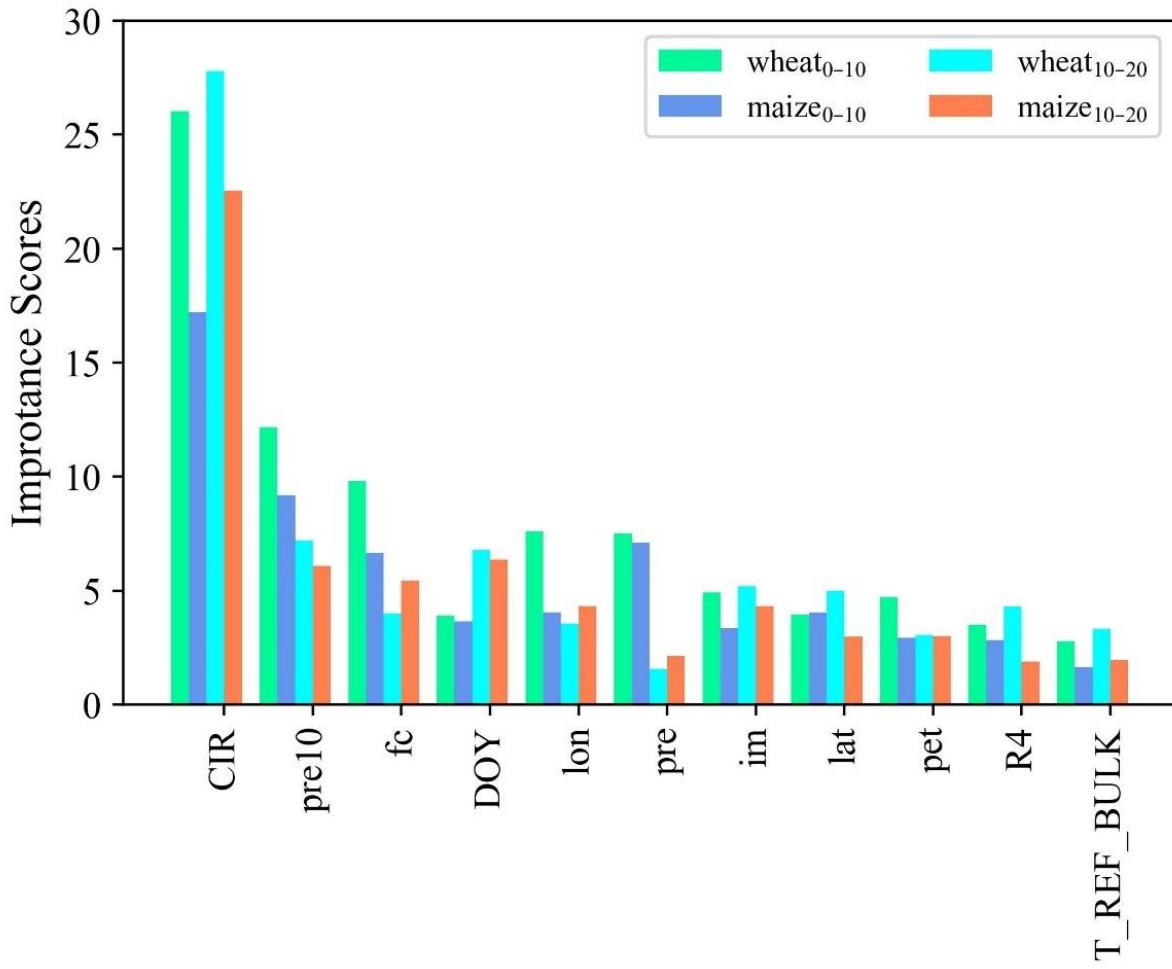


**Figure 5 The importance scores of 11 independent variables and irrigation mode (CIR).**





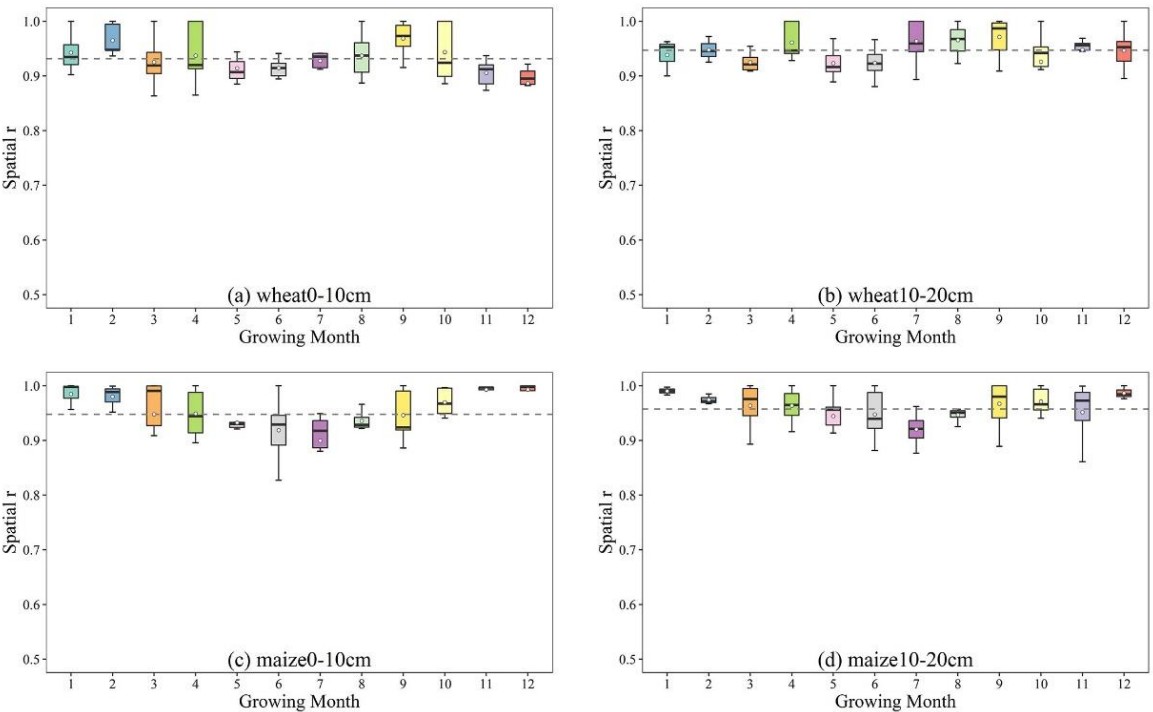

**Figure 6 Comparison of the spatial accuracy (*r*) between ChinaCropSM1km and in situ SM observations in each month by crops and depths. (a) wheat$_{0-10}$, (b) wheat$_{10-20}$, (c) maize$_{0-10}$ and (d) maize$_{10-20}$. The dash lines represent the mean values.**



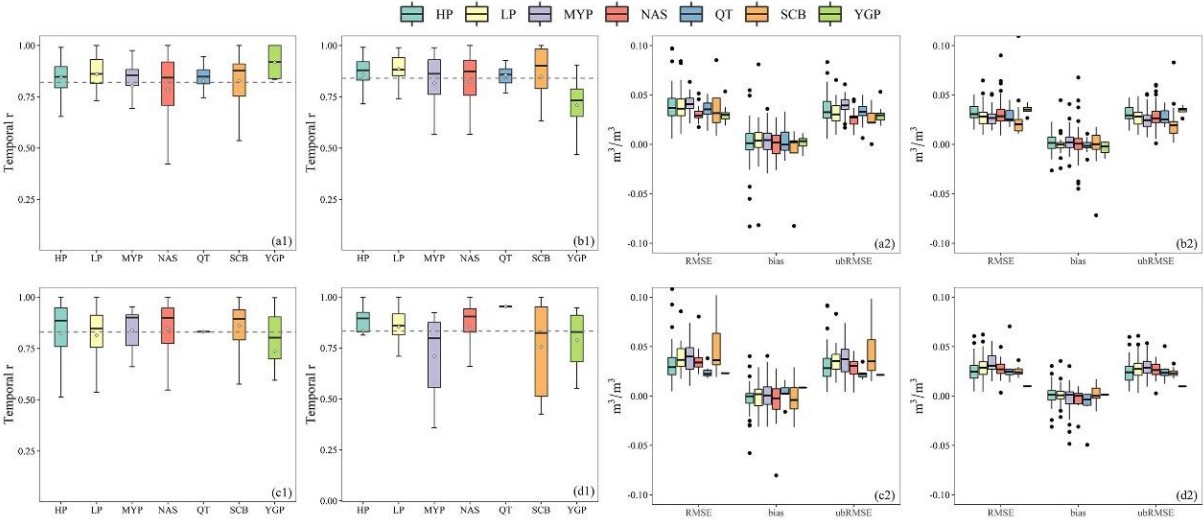

**Figure 7 Comparison of the temporal accuracy (*r*, RMSE, bias, ubRMSE) between ChinaCropSM1km and in situ soil moisture observations by crops and depths. (a1, a2) wheat$_{0-10}$, (b1, b2) wheat$_{10-20}$, (c1, c2) maize$_{0-10}$ and (d1, d2) maize$_{10-20}$. The dash lines represent the mean values.**






**Table 3 Summary on means of evaluation indexes (*r*, bias, RMSE, and ubRMSE) of three products (ChinaCropSM1km, RSSSM and ESA CCI SM), all products were compared with in situ surface observations (0–10 cm).**

| Product | ChinaCrop SM1km$_{maize}$ | RSSSM | ESA CCI SM | ChinaCrop SM1km$_{wheat}$ | RSSSM | ESA CCI SM |
|---|---|---|---|---|---|---|
| *r* | **0.93** | 0.43 | 0.35 | **0.93** | 0.29 | 0.33 |
| RMSE | **0.033** | 0.167 | 0.126 | **0.035** | 0.187 | 0.121 |
| bias | **0.0006** | −0.1361 | −0.0846 | **−0.0008** | −0.1552 | −0.0705 |
| ubRMSE | **0.033** | 0.097 | 0.093 | **0.035** | 0.105 | 0.099 |





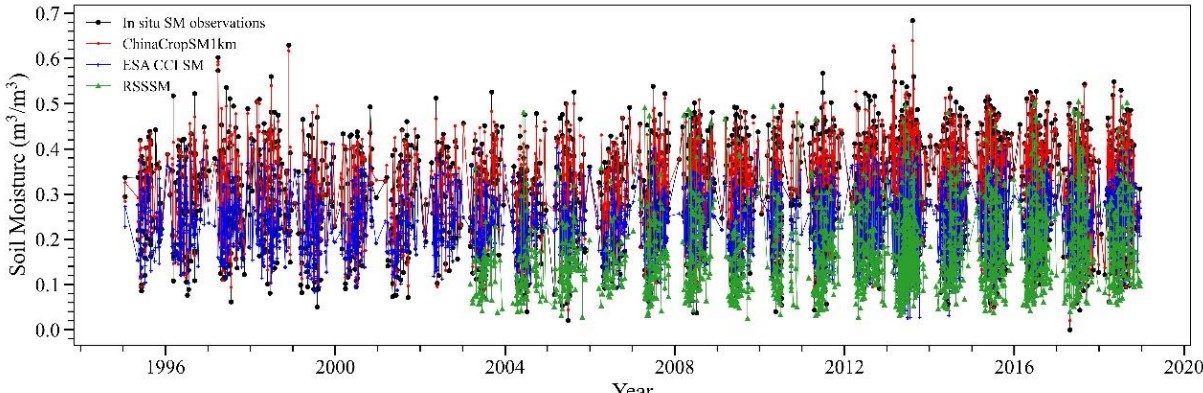

**Figure 8 Time series of comparison between in situ SM observations and products.**




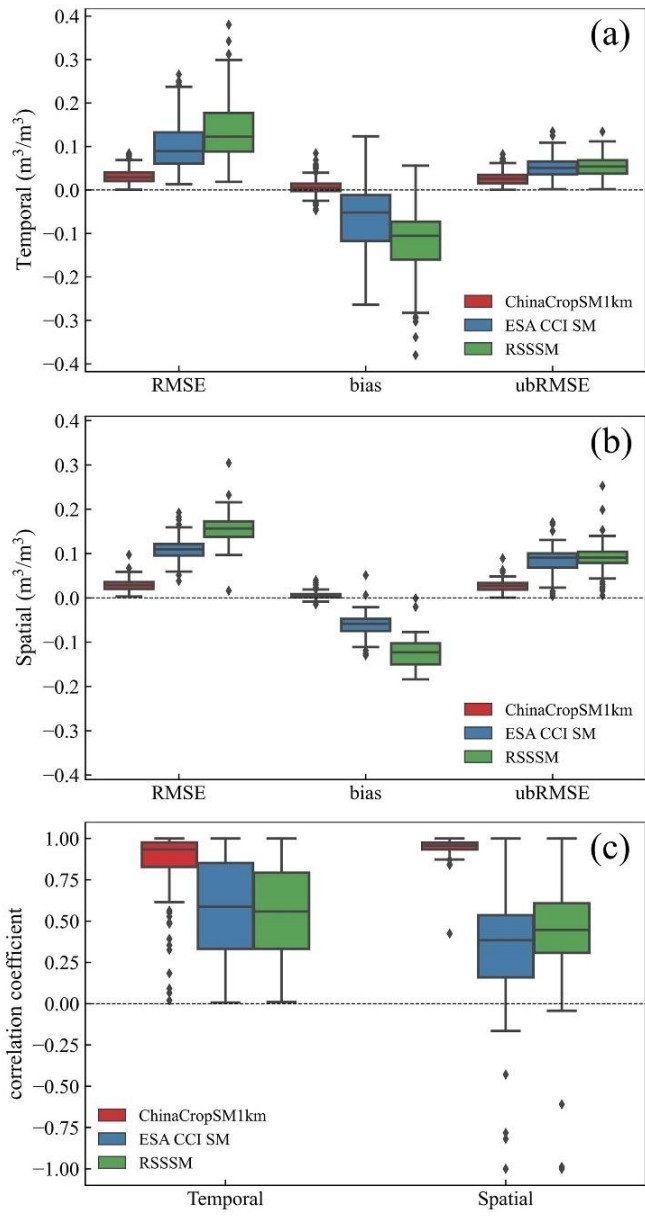

**Figure 9 Boxplot of the temporal (a, c) and spatial (b, c) accuracy for ChinaCropSM1km, RSSSM and ESA CCI SM by *r*, bias, RMSE, and ubRMSE. These evaluation indexes were calculated by comparing the three products with in situ SM observations; the comparison period for ChinaCropSM1km and RSSSM was from 2003 to 2018; and for ChinaCropSM1km and ESA CCI SM was 1995−2018.**
