# Peer review of "ChinaCropSM1 km: a fine 1 km daily Soil Moisture dataset for dryland wheat and maize across China during 1993–2018"

_Earth System Science Data, 2022_

## Author Comment (AC2)

**Dear Editor and Reviewer # 1:**

Thanks for your careful reviewing and all constructive comments on our manuscript. We have taken all your comments into account and responded positively to qualify our manuscript for a potential publication in the journal. Our responses are written in blue below.

Comment 1:

I was very impressed by such valuable daily SM for more than 20 years over the whole mainland of China. Comparing with quantities of public products retrieved from remote sensing or downscaling into fine resolution, Chinacropland really open a new window for us to provide key parameters on earth observations. Irrigation practices do play more significances on crop production in China, especially for dryland crop. Therefore, no any doubt will be shown on the values of irrigation sub-model. Such novelty imply a potential way for applying irrigation sub-model into other areas and crops in the world. The study is fallen closely within the scope of ESSD. However, the authors should consider my several concerns below before their submission being accepted.

Thank for your positive comments, which really encourage us to improve our study.

(1) I am wondering how they obtain the crop dryland maps. For wheat or maize, it seem to me the location is constant. I need more detailed information to better understand their study.

**Response:**

Yes, we did remain the ChinaCropland location constant as several publications did similarly (Gervois et al., 2008; Ke et al., 2018). We proposed a new crop phenology-based crop mapping approach to generate a 1 km harvesting area dataset for three staple crops (i.e. rice, wheat, and maize) in China from 2000 to 2015 based on GLASS leaf area index (LAI) products (Luo et al., 2020a, b). Actually, we used the union of the annual harvested area dataset for maize and wheat as the China crop drylands maps.

Reference:

Gervois, S., Ciais, P., de Noblet-Ducoudré, N., Brisson, N., Vuichard, N., and Viovy, N.: Carbon and water balance of European croplands throughout the 20th century: CARBON BALANCE OF EUROPEAN CROPLANDS, Global Biogeochem. Cycles, 22, n/a-n/a,

https://doi.org/10.1029/2007GB003018, 2008.

Ke, X., van Vliet, J., Zhou, T., Verburg, P. H., Zheng, W., and Liu, X.: Direct and indirect loss of natural habitat due to built-up area expansion: A model-based analysis for the city of Wuhan, China, Land Use Policy, 74, 231–239, https://doi.org/10.1016/j.landusepol.2017.12.048, 2018.

Luo, Y., Zhang, Z., Chen, Y., Li, Z., and Tao, F.: ChinaCropPhen1km: a high-resolution crop phenological dataset for three staple crops in China during 2000–2015 based on leaf area index (LAI) products, Earth Syst. Sci. Data, 12, 197–214, https://doi.org/10.5194/essd-12-197-2020, 2020a.

Luo, Y., Zhang, Z., Li, Z., Chen, Y., Zhang, L., Cao, J., and Tao, F.: Identifying the spatiotemporal changes of annual harvesting areas for three staple crops in China by integrating multi-data sources, Environ. Res. Lett., 15, 074003, https://doi.org/10.1088/1748-9326/ab80f0, 2020b.

(2) I do not think RF is a new method to retrieve SM. That is to say, more interesting findings have ascribed from combining irrigation module into SM estimation model. However, the authors have not specified the point. I am looking forward to more information on it, e.g. the accuracy comparison between with irrigation module and without it.

**Response:** Many thanks for your valuable suggestion. We have compared their accuracy results between with-irrigation module and without-irrigation module as supplemental materials. Please see it in **Table S5**. The improved accuracy results were consistently indicated by all comparisons, e.g. decreases in RMSE, and ubRMSE and increase in $R^2$.

**Table S5** The accuracy comparison between with irrigation module (in bold) and without it.

| ChinaCropSM1km | BIAS | | $R^2$ | | RMSE | | ubRMSE | |
|---|---|---|---|---|---|---|---|---|
| wheat$_{0-10}$ | **−0.0011** | −0.0019 | **0.860** | 0.801 | **0.037** | 0.044 | **0.037** | 0.044 |
| wheat$_{10-20}$ | **−0.0002** | −0.0006 | **0.895** | 0.838 | **0.031** | 0.039 | **0.031** | 0.039 |
| maize$_{0-10}$ | **0.0009** | 0.0007 | **0.861** | 0.798 | **0.036** | 0.043 | **0.036** | 0.043 |
| maize$_{10-20}$ | **0.0003** | −0.0001 | **0.894** | 0.812 | **0.029** | 0.038 | **0.029** | 0.038 |

(3) Deeper and more extent discussions will further expand the reputation and influence of their

study.

**Response:** Thanks very much for your constructive comment. We have followed you to insert deeper and more extent discussions into our manuscript (Line 307~320 in the revised manuscript).

"The ChinaCropSM dataset are credible and accurate according to the results comparing with the public datasets, however, some limitations are still existed in our study. First, the limited AMS irrigation records may lead to the uncertainty in the irrigation factor predictions. More detailed irrigation information will help to improve irrigation module performances. Second, our method for generating cropland SM is applicable to other regions and crops, but more environmental variables will be increasingly required considering the SM variabilities are complex processes controlled by many factors (Famiglietti et al., 2008; Qin et al., 2013; Guevara and Vargas, 2019), especially for irrigation activities. For example, to characterize more accurately the irrigation activities, many field samples are highly required in both spatial and temporal resolutions. Other auxiliary data on information of crop growth, classification, and managements (e.g. irrigation frequency, amount and method) will benefit to develop our irrigation module and derive SM datasets more accurately. Moreover, advanced algorithms will be potential alternatives for random forest due to its strong dependence on inputs (Breiman, 2001; Rasmussen, 2004). Improving irrigation module should be focused on details such as irrigation amount and frequency, which will significantly help to verify and improve the accuracy of both irrigation and SM predictions.".

(4) Generally, the English writing is Ok. But typo can be observed sometimes, a careful check should be conducted throughout their manuscript.

**Response:** Thank you for your careful comments. We have modified carefully throughout the revised paper (Line 58, 100, 143, 524).

Line 58: $r^2$ -> $R^2$

Line 100: "accumulated precipitation for 10 days" -> "ante-accumulated precipitation over ten days"

Line 143:

Line 524: "mode" -> "factor"

---

## Author Comment (AC3)

**Dear Editor and Reviewer # 2:**

This study provides a longer term soil moisture dataset (ChinaCropSM1km) for crop drylands across mainland of China. ChinaCropSM1km perform better than public product in both higher accuracy and more details (daily, more soil layers) by using machine learning technology. Such soil moisture dataset with higher resolutions is very valuable for the studies on crop model, yield estimation, and climate change impact assessment. Moreover, their methodology is robust, and their interesting results were well interpreted. The irrigation module is a novel way to improve highly moisture estimation. Therefore, I recommend it can be accepted after a minor revision.

We appreciate your insightful comments on our paper. The comments offered have been immensely helpful. We have responded to every question, indicating exactly how we addressed each concern or problem and describing the changes we have made. The revisions have been approved by all authors. The point-to-point responses to your comments are listed below in **blue**.

**Comments and suggestions:**

**Point 1:** There is a problem with the resolution. The ground observation data is point measurement data, how to match the resolution of 1km? Please explain this in the manuscript.

**Response:** Thank you a lot for the insightful suggestion. We have followed you to insert relevant contents into our manuscript (highlighted in "Track Changes", line 140~143, 185~186).

"We use the Extract Values to Points tool to extract the 1km resolution raster information of environmental (i.e. SP, RSD and GI) data to AMS point data, output point data attributes and save it in CSV format to obtain a data set of environmental factors through ArcGIS 10.5.".

"All these point samples are used to develop pointed-SM model, and then applied these pointed-models developed to inversely calculate the gridded-SM by inputting 1km-raster environmental variables.".

**Point 2:** Section 2.1. The authors pointed out that the study area is dominated by dryland crops (i.e.

wheat and maize) in China, how was the Chinacropland layer defined in Figure 1 according to the annual crop harvested area in mainland China from 2000 to 2015? please describe the details.

**Response:**

We remain the ChinaCropland location constant as several publications did similarly (Gervois et al., 2008; Ke et al., 2018). We proposed a new crop phenology-based crop mapping approach to generate a 1 km harvesting area dataset for three staple crops (i.e. rice, wheat, and maize) in China from 2000 to 2015 based on GLASS leaf area index (LAI) products (Luo et al., 2020a, b). Actually, we used the union of the annual harvested area dataset for maize and wheat as the China crop drylands maps.

Reference:

Gervois, S., Ciais, P., de Noblet-Ducoudré, N., Brisson, N., Vuichard, N., and Viovy, N.: Carbon and water balance of European croplands throughout the 20th century: CARBON BALANCE OF EUROPEAN CROPLANDS, Global Biogeochem. Cycles, 22, n/a-n/a, https://doi.org/10.1029/2007GB003018, 2008.

Ke, X., van Vliet, J., Zhou, T., Verburg, P. H., Zheng, W., and Liu, X.: Direct and indirect loss of natural habitat due to built-up area expansion: A model-based analysis for the city of Wuhan, China, Land Use Policy, 74, 231–239, https://doi.org/10.1016/j.landusepol.2017.12.048, 2018.

Luo, Y., Zhang, Z., Chen, Y., Li, Z., and Tao, F.: ChinaCropPhen1km: a high-resolution crop phenological dataset for three staple crops in China during 2000–2015 based on leaf area index (LAI) products, Earth Syst. Sci. Data, 12, 197–214, https://doi.org/10.5194/essd-12-197-2020, 2020a.

Luo, Y., Zhang, Z., Li, Z., Chen, Y., Zhang, L., Cao, J., and Tao, F.: Identifying the spatiotemporal changes of annual harvesting areas for three staple crops in China by integrating multi-data sources, Environ. Res. Lett., 15, 074003, https://doi.org/10.1088/1748-9326/ab80f0, 2020b.

**Point 3:** In (1), the author judges the irrigation factors by comparing the observed soil moisture and the soil moisture evaluation index (SMI) according to the corresponding soil depth and phenology of crops. However, I notice that the SMI in Table 2 is a range, rather than an exact number. Please give reasonable explanation for this.

**Response:**

Actually, we use the minimum value of the SMI interval (i.e. an exact number threshold) to judge

the irrigation factors considering the spatial differences in irrigated cropland. The irrigation factor (CIR) is assigned by 1 if the actual soil moisture is larger than the irrigation threshold. We used the minimum value to ensure that CIR were taken into account in all zones during forecasting SM. Using the minimum value might misclassify CIR, such as assigning "1" to no irrigation application, but such treatment is a compromise way before more detailed irrigation information is available. Moreover, we explained this limitation in the discussion section.

**Point 4:** In section 2.3.2, considering the new SM product has been derived by integrating the irrigation module into SM model, it is better to evaluate accuracy of the module (irrigation factor forecasting model) and supply such important information into new edition.

**Response:** In the Data and methods (Section 2.2.1), the accuracy of irrigation factor forecasting model has been provided in the revised manuscript (Line 214~220).

We evaluated our irrigation factor forecasting model results (**Table S4**) using the receiver operating characteristics (ROC) curve and their Area Under the Curve (AUC) (Fawcett, 2006). Also, we calculated UA (Eq. 7), PA (Eq. 8), and overall accuracy (Eq. 9) based on confusion matrices (**Table S3**) containing the percentages of the four possible outcomes of a model: True Positive (TP), True Negative (TN), False Positive (FP) and False Negative (FN) (Fawcett, 2006).

$$PA = \frac{TP}{TP+FP} \tag{7}$$

$$UA = \frac{TP}{TP+FN} \tag{8}$$

$$Accuracy = \frac{TP+TN}{TP+FP+TN+FN} \tag{9}$$

Reference:

Fawcett, T.: An introduction to ROC analysis, Pattern Recognition Letters, 27, 861–874, https://doi.org/10.1016/j.patrec.2005.10.010, 2006.

Table S3 Confusion matrix table in this study.

| | | Class | |
| --- | --- | --- | --- |
| | | Irrigated | Non |
| Reference | Irrigated | TP | FN |
| | Non | FP | TN |

Table S4 Confusion matrix of irrigated validation based on the test dataset. Prediction categories are columns while reference categories are rows.

| ChinaCropSM1km | Class | Irrigated | Non | Total | Accuracy | PA | UA | AUC |
| --- | --- | --- | --- | --- | --- | --- | --- | --- |
| $wheat_{0-10}$ | Irrigated | 1633 | 395 | 2028 | 0.85 | 0.82 | 0.81 | 0.84 |
| | Non | 365 | 2744 | 3109 | | | | |
| | Total | 1998 | 3139 | | | | | |
| $wheat_{10-20}$ | Irrigated | 1583 | 446 | 2029 | 0.84 | 0.81 | 0.78 | 0.83 |
| | Non | 365 | 2749 | 3114 | | | | |
| | Total | 1948 | 3195 | | | | | |
| $maize_{0-10}$ | Irrigated | 915 | 310 | 1225 | 0.86 | 0.85 | 0.75 | 0.84 |
| | Non | 167 | 2030 | 2197 | | | | |
| | Total | 1082 | 2340 | | | | | |
| $maize_{10-20}$ | Irrigated | 875 | 321 | 1196 | 0.86 | 0.83 | 0.73 | 0.83 |
| | Non | 175 | 2052 | 2227 | | | | |
| | Total | 1050 | 2373 | | | | | |

**Point 5:** Some typos are found in manuscript, and check manuscript carefully and correct them. e.g. Line143: delete 'in China'.

**Response:** Thank you for your careful comments. We have modified all typos in the revised paper (Line 58, 100, 143, 524).

Line 58: $r^2$ -> $R^2$

Line 100: "accumulated precipitation for 10 days" -> "ante-accumulated precipitation over ten days"

Line 143:

Line 524: "mode" -> "factor"

**Point 6:** Figure 2 should be improved. Currently, some labels are too vague to clearly identify.

**Response:** Many thanks for your advice. We have modified it in the revised paper.

**Point 7:** Please modify the line widths in Table 2.

**Response:** Many thanks for your careful check. We have modified it in the revised paper.

**Point 8:** Line257: insert blank between two words. 'Figure8' -> 'Figure 8'.

**Response:** Thanks for your careful review. We have modified it in the revised paper. (Line 284).

**Point 9:** Figure S5 was not used in the main text, please cite it in main text or delete it from supplemental material.

**Response:** Thanks for your careful review. We have deleted it from supplemental material.

---

## Author Comment (AC4)

**Dear Editor and Reviewer # 3:**

We appreciate your insightful comments on our paper. The comments offered have been immensely helpful. We have responded to every question, indicating exactly how we addressed each concern or problem and describing the changes we have made. The revisions have been approved by all authors. The point-to-point responses to your comments are listed below in **blue**.

This is an interesting effort in developing the SM product for crop dryland, which has potential for various applications. The paper is well written and organized. Taking the CIR as a predictor seems to be a useful way to predict SM in crop dryland. However, I have some concerns as following. Please pay more attention on the comments about line 174-175.

**Point 1:** Why only mapping SM for dryland, not rice?

**Response:** Rice is commonly grown in southern areas with plenty rainfall or northern areas well equipped by irrigation in China. Therefore, soil moisture is usually over saturated and keeps constant (near 100%) during the whole growing season (Zheng et al., 2000; Alhaj Hamoud et al., 2019). Considering the significant role of SM for maize and wheat planted in dryland across China, we mapped the SM for crop drylands, not including rice.

Reference:
Alhaj Hamoud, Y., Guo, X., Wang, Z., Shaghaleh, H., Chen, S., Hassan, A., and Bakour, A.: Effects of irrigation regime and soil clay content and their interaction on the biological yield, nitrogen uptake and nitrogen-use efficiency of rice grown in southern China, Agricultural Water Management, 213, 934–946, https://doi.org/10.1016/j.agwat.2018.12.017, 2019.

Zheng, X., Wang, M., Wang, Y., Shen, R., Gou, J., Li, J., Jin, J., and Li, L.: Impacts of soil moisture on nitrous oxide emission from croplands: a case study on the rice-based agro-ecosystem in Southeast China, Chemosphere - Global Change Science, 2, 207–224, https://doi.org/10.1016/S1465-9972(99)00056-2, 2000.

**Point 2:** Line 110-115: there are two sources of FC which one is used?

**Response:** Field capacity (fc) was obtained from OpenLandMap which included fc under 33kPa at 0cm (b0) and 10 cm (b10) depth. When predict $ChinaCropSM_{0-10cm}$, we used fc under 33kPa at 0cm (b0) depth. When predict $ChinaCropSM_{10-20cm}$, we used fc under 33kPa at 10cm (b10) depth.

**Point 3:** Line 120: the short name "AMS" is used only one time. Consider full name. In addition, what is R4, R5 and R16? And it should not be calculated only for AMS but for each cell, as a predictor.

**Response:** Many thanks for your advice. We have increased the full name of "AMS" in the revised paper.

Yes, R4, R5 and R16 is calculated for each cell, as a predictor. Actually, the R4, R5 and R16 are river network vector data at different levels in China. When training sample data, we calculate the distance for AMS. Additionally, we calculated the distance from each cell to river network vector data when predicting the ChinaCropSM.

**Point 4:** Line 171: Grammar error. Not a complete sentence.

**Response:** Thank you for your careful comments. We have modified it (Line 180).

"As for the response variable (Classified Irrigation CIR), it is calculated by irrigation threshold (Table 2) and in situ information, including crop type, phenology and soil depth.".

**Point 5:** Line 174-175: It should not be random splitting because SM of different time from the same site may be highly correlated. This will give a higher performance for the model. Instead, the splitting should be based on sites, i.e., data from a site should be all in the training set or all in test set. Note that the model is predicting unknown locations based on the observing sites, and the spatial interpolation ability should be evaluated by the site-based splitting.

**Response:** Thanks very much for your constructive comment.

According to your site-based splitting method, we re-optimized the hyper-parameters of the prediction model to reduce overfitting and evaluated the prediction results. We found the soil moisture predicted by your method agreed well with in situ SM observations (ubRMSE ranges from 0.046–0.057, and $R^2$ ranges from 0.642–0.761), although the model performance drops slightly (Figure 1).

Similarly, in the case of site-based splitting, all prediction accuracy of SM were consistently improved both for crops and depths with comparison of those without an irrigation module (e.g. $R^2$ increased by 9–41%, ubRMSE decreased by 21–26%) (Figure 2). Also, we further compared our ChinaCropSM1km with the two popular public global SM products (Table 1). All indexes of our ChinaCropSM were consistently indicated by the higher accuracy.

Different splitting methods during training and testing do affect model performance. Selecting which splitting method to improve the generalization performance is dependent on data. Generally, the larger size of data, the smaller effect of the splitting methods on the results (Birba, 2020). Therefore, the model performances of two splitting methods show no significant differences because of quantities of field observations available in our study. We have followed you to insert deeper and more extent discussions into our manuscript (Line 307~322 in the revised manuscript).

Reference:

Birba, D. E.: A Comparative study of data splitting algorithms for machine learning model selection, 2020.

The results are following:

[Figure]

Figure 1 Comparison between the predicted soil moisture (ChinaCropSM1km) and in situ samples by crops and depths (cm) according to site-based splitting.

[Figure]

Figure 2 Comparison of soil moisture accuracy between with irrigation and without an irrigation module according to site-based splitting.

Table 1 Summary on means of evaluation indexes of three products (ChinaCropSM1km, RSSSM and ESA CCI SM).

| Product | ChinaCropSM | RSSSM | ESA CCI SM |
|---------|-------------|-------|------------|
| $r$ | 0.85 | 0.52 | 0.42 |
| RMSE | 0.054 | 0.144 | 0.120 |
| bias | –0.005 | –0.112 | –0.066 |
| ubRMSE | 0.054 | 0.092 | 0.100 |

**Point 6:** Line 185: How many times do you run the model to get the importance, as the importance will be different each time. It should take the average importance of dozens of runs like 100.

**Response:** Yes, we did take the averages of dozens of runs. We ran each training model 50 times to get the importance and averaged the importance outcome.

**Point 7:** Fig.6 and 7: what are the different boxes stand for?

**Response:** The boxes in Fig.6 and Fig.7 actually stand for different results, with spatial pattern in Fig.6 and temporal one in Fig. 7. Both patterns were conducted between ChinaCropSM1km and the in situ SM observations.

The horizontal line within each box stands for median, the white dot for mean, the box bottom for first quantile, the top for third quantile, and black dots for outliers.

**Point 8:** Section 3.5: I do not think this comparison is fare. The evaluation using the test data for Cropland should be used instead of all in situ data because the model used them to establish leading to an independent evaluation.

**Response:** Actually, we only used the testing data for evaluating, not including all in situ data. We agreed well with you that using all observations will lead to an independent evaluation.

---

## Author Response (AR3)

**Dear Editor and Reviewer # 1:**

Thanks for your careful reviewing and all constructive comments on our manuscript. We have taken all your comments into account and responded positively to qualify our manuscript for a potential publication in the journal. Our responses are written in blue below.

Comment 1:

I was very impressed by such valuable daily SM for more than 20 years over the whole mainland of China. Comparing with quantities of public products retrieved from remote sensing or downscaling into fine resolution, Chinacropland really open a new window for us to provide key parameters on earth observations. Irrigation practices do play more significances on crop production in China, especially for dryland crop. Therefore, no any doubt will be shown on the values of irrigation sub-model. Such novelty imply a potential way for applying irrigation sub-model into other areas and crops in the world. The study is fallen closely within the scope of ESSD. However, the authors should consider my several concerns below before their submission being accepted.

Thank for your positive comments, which really encourage us to improve our study.

(1) I am wondering how they obtain the crop dryland maps. For wheat or maize, it seem to me the location is constant. I need more detailed information to better understand their study.

**Response:**

Yes, we did remain the ChinaCropland location constant as several publications did similarly (Gervois et al., 2008; Ke et al., 2018). We proposed a new crop phenology-based crop mapping approach to generate a 1 km harvesting area dataset for three staple crops (i.e. rice, wheat, and maize) in China from 2000 to 2015 based on GLASS leaf area index (LAI) products (Luo et al., 2020a, b). Actually, we used the union of the annual harvested area dataset for maize and wheat as the China crop drylands maps.

Reference:

Gervois, S., Ciais, P., de Noblet-Ducoudré, N., Brisson, N., Vuichard, N., and Viovy, N.: Carbon and water balance of European croplands throughout the 20th century: CARBON BALANCE OF EUROPEAN CROPLANDS, Global Biogeochem. Cycles, 22, n/a-n/a,

https://doi.org/10.1029/2007GB003018, 2008.

Ke, X., van Vliet, J., Zhou, T., Verburg, P. H., Zheng, W., and Liu, X.: Direct and indirect loss of natural habitat due to built-up area expansion: A model-based analysis for the city of Wuhan, China, Land Use Policy, 74, 231–239, https://doi.org/10.1016/j.landusepol.2017.12.048, 2018.

Luo, Y., Zhang, Z., Chen, Y., Li, Z., and Tao, F.: ChinaCropPhen1km: a high-resolution crop phenological dataset for three staple crops in China during 2000–2015 based on leaf area index (LAI) products, Earth Syst. Sci. Data, 12, 197–214, https://doi.org/10.5194/essd-12-197-2020, 2020a.

Luo, Y., Zhang, Z., Li, Z., Chen, Y., Zhang, L., Cao, J., and Tao, F.: Identifying the spatiotemporal changes of annual harvesting areas for three staple crops in China by integrating multi-data sources, Environ. Res. Lett., 15, 074003, https://doi.org/10.1088/1748-9326/ab80f0, 2020b.

(2) I do not think RF is a new method to retrieve SM. That is to say, more interesting findings have ascribed from combining irrigation module into SM estimation model. However, the authors have not specified the point. I am looking forward to more information on it, e.g. the accuracy comparison between with irrigation module and without it.

**Response:** Many thanks for your valuable suggestion. We have compared their accuracy results between with-irrigation module and without-irrigation module as supplemental materials. Please see it in **Table S5**. The improved accuracy results were consistently indicated by all comparisons, e.g. decreases in RMSE, and ubRMSE and increase in $R^2$.

**Table S5** The accuracy comparison between with irrigation module (in bold) and without it.

| ChinaCropSM1km | BIAS | | $R^2$ | | RMSE | | ubRMSE | |
|---|---|---|---|---|---|---|---|---|
| $wheat_{0-10}$ | **−0.0011** | −0.0019 | **0.860** | 0.801 | **0.037** | 0.044 | **0.037** | 0.044 |
| $wheat_{10-20}$ | **−0.0002** | −0.0006 | **0.895** | 0.838 | **0.031** | 0.039 | **0.031** | 0.039 |
| $maize_{0-10}$ | **0.0009** | 0.0007 | **0.861** | 0.798 | **0.036** | 0.043 | **0.036** | 0.043 |
| $maize_{10-20}$ | **0.0003** | −0.0001 | **0.894** | 0.812 | **0.029** | 0.038 | **0.029** | 0.038 |

(3) Deeper and more extent discussions will further expand the reputation and influence of their

study.

**Response:** Thanks very much for your constructive comment. We have followed you to insert deeper and more extent discussions into our manuscript (Line 307~320 in the revised manuscript).

"The ChinaCropSM dataset are credible and accurate according to the results comparing with the public datasets, however, some limitations are still existed in our study. First, the limited AMS irrigation records may lead to the uncertainty in the irrigation factor predictions. More detailed irrigation information will help to improve irrigation module performances. Second, our method for generating cropland SM is applicable to other regions and crops, but more environmental variables will be increasingly required considering the SM variabilities are complex processes controlled by many factors (Famiglietti et al., 2008; Qin et al., 2013; Guevara and Vargas, 2019), especially for irrigation activities. For example, to characterize more accurately the irrigation activities, many field samples are highly required in both spatial and temporal resolutions. Other auxiliary data on information of crop growth, classification, and managements (e.g. irrigation frequency, amount and method) will benefit to develop our irrigation module and derive SM datasets more accurately. Moreover, advanced algorithms will be potential alternatives for random forest due to its strong dependence on inputs (Breiman, 2001; Rasmussen, 2004). Improving irrigation module should be focused on details such as irrigation amount and frequency, which will significantly help to verify and improve the accuracy of both irrigation and SM predictions.".

(4) Generally, the English writing is Ok. But typo can be observed sometimes, a careful check should be conducted throughout their manuscript.

**Response:** Thank you for your careful comments. We have modified carefully throughout the revised paper (Line 58, 100, 143, 524).

Line 58: $r^2$ -> $R^2$

Line 100: "accumulated precipitation for 10 days" -> "ante-accumulated precipitation over ten days"

Line 143:

Line 524: "mode" -> "factor"

**Dear Editor and Reviewer # 2:**

This study provides a longer term soil moisture dataset (ChinaCropSM1km) for crop drylands across mainland of China. ChinaCropSM1km perform better than public product in both higher accuracy and more details (daily, more soil layers) by using machine learning technology. Such soil moisture dataset with higher resolutions is very valuable for the studies on crop model, yield estimation, and climate change impact assessment. Moreover, their methodology is robust, and their interesting results were well interpreted. The irrigation module is a novel way to improve highly moisture estimation. Therefore, I recommend it can be accepted after a minor revision.

We appreciate your insightful comments on our paper. The comments offered have been immensely helpful. We have responded to every question, indicating exactly how we addressed each concern or problem and describing the changes we have made. The revisions have been approved by all authors. The point-to-point responses to your comments are listed below in **blue**.

**Comments and suggestions:**

**Point 1:** There is a problem with the resolution. The ground observation data is point measurement data, how to match the resolution of 1km? Please explain this in the manuscript.

**Response:** Thank you a lot for the insightful suggestion. We have followed you to insert relevant contents into our manuscript (highlighted in "Track Changes", line 140~143, 185~186).

"We use the Extract Values to Points tool to extract the 1km resolution raster information of environmental (i.e. SP, RSD and GI) data to AMS point data, output point data attributes and save it in CSV format to obtain a data set of environmental factors through ArcGIS 10.5.".

"All these point samples are used to develop pointed-SM model, and then applied these pointed-models developed to inversely calculate the gridded-SM by inputting 1km-raster environmental variables.".

**Point 2:** Section 2.1. The authors pointed out that the study area is dominated by dryland crops (i.e.

wheat and maize) in China, how was the Chinacropland layer defined in Figure 1 according to the annual crop harvested area in mainland China from 2000 to 2015? please describe the details.

**Response:**

We remain the ChinaCropland location constant as several publications did similarly (Gervois et al., 2008; Ke et al., 2018). We proposed a new crop phenology-based crop mapping approach to generate a 1 km harvesting area dataset for three staple crops (i.e. rice, wheat, and maize) in China from 2000 to 2015 based on GLASS leaf area index (LAI) products (Luo et al., 2020a, b). Actually, we used the union of the annual harvested area dataset for maize and wheat as the China crop drylands maps.

Reference:

Gervois, S., Ciais, P., de Noblet-Ducoudré, N., Brisson, N., Vuichard, N., and Viovy, N.: Carbon and water balance of European croplands throughout the 20th century: CARBON BALANCE OF EUROPEAN CROPLANDS, Global Biogeochem. Cycles, 22, n/a-n/a, https://doi.org/10.1029/2007GB003018, 2008.

Ke, X., van Vliet, J., Zhou, T., Verburg, P. H., Zheng, W., and Liu, X.: Direct and indirect loss of natural habitat due to built-up area expansion: A model-based analysis for the city of Wuhan, China, Land Use Policy, 74, 231–239, https://doi.org/10.1016/j.landusepol.2017.12.048, 2018.

Luo, Y., Zhang, Z., Chen, Y., Li, Z., and Tao, F.: ChinaCropPhen1km: a high-resolution crop phenological dataset for three staple crops in China during 2000–2015 based on leaf area index (LAI) products, Earth Syst. Sci. Data, 12, 197–214, https://doi.org/10.5194/essd-12-197-2020, 2020a.

Luo, Y., Zhang, Z., Li, Z., Chen, Y., Zhang, L., Cao, J., and Tao, F.: Identifying the spatiotemporal changes of annual harvesting areas for three staple crops in China by integrating multi-data sources, Environ. Res. Lett., 15, 074003, https://doi.org/10.1088/1748-9326/ab80f0, 2020b.

**Point 3:** In (1), the author judges the irrigation factors by comparing the observed soil moisture and the soil moisture evaluation index (SMI) according to the corresponding soil depth and phenology of crops. However, I notice that the SMI in Table 2 is a range, rather than an exact number. Please give reasonable explanation for this.

**Response:**

Actually, we use the minimum value of the SMI interval (i.e. an exact number threshold) to judge

the irrigation factors considering the spatial differences in irrigated cropland. The irrigation factor (CIR) is assigned by 1 if the actual soil moisture is larger than the irrigation threshold. We used the minimum value to ensure that CIR were taken into account in all zones during forecasting SM. Using the minimum value might misclassify CIR, such as assigning "1" to no irrigation application, but such treatment is a compromise way before more detailed irrigation information is available. Moreover, we explained this limitation in the discussion section.

**Point 4:** In section 2.3.2, considering the new SM product has been derived by integrating the irrigation module into SM model, it is better to evaluate accuracy of the module (irrigation factor forecasting model) and supply such important information into new edition.

**Response:** In the Data and methods (Section 2.2.1), the accuracy of irrigation factor forecasting model has been provided in the revised manuscript (Line 214~220).

We evaluated our irrigation factor forecasting model results (**Table S4**) using the receiver operating characteristics (ROC) curve and their Area Under the Curve (AUC) (Fawcett, 2006). Also, we calculated UA (Eq. 7), PA (Eq. 8), and overall accuracy (Eq. 9) based on confusion matrices (**Table S3**) containing the percentages of the four possible outcomes of a model: True Positive (TP), True Negative (TN), False Positive (FP) and False Negative (FN) (Fawcett, 2006).

$$PA = \frac{TP}{TP+FP} \tag{7}$$

$$UA = \frac{TP}{TP+FN} \tag{8}$$

$$Accuracy = \frac{TP+TN}{TP+FP+TN+FN} \tag{9}$$

Reference:

Fawcett, T.: An introduction to ROC analysis, Pattern Recognition Letters, 27, 861–874, https://doi.org/10.1016/j.patrec.2005.10.010, 2006.

Table S3 Confusion matrix table in this study.

| | | Class | |
|---|---|---|---|
| | | Irrigated | Non |
| Reference | Irrigated | TP | FN |
| | Non | FP | TN |

Table S4 Confusion matrix of irrigated validation based on the test dataset. Prediction categories are columns while reference categories are rows.

| ChinaCropSM1km | Class | Irrigated | Non | Total | Accuracy | PA | UA | AUC |
|---|---|---|---|---|---|---|---|---|
| $\text{wheat}_{0-10}$ | Irrigated | 1633 | 395 | 2028 | 0.85 | 0.82 | 0.81 | 0.84 |
| | Non | 365 | 2744 | 3109 | | | | |
| | Total | 1998 | 3139 | | | | | |
| $\text{wheat}_{10-20}$ | Irrigated | 1583 | 446 | 2029 | 0.84 | 0.81 | 0.78 | 0.83 |
| | Non | 365 | 2749 | 3114 | | | | |
| | Total | 1948 | 3195 | | | | | |
| $\text{maize}_{0-10}$ | Irrigated | 915 | 310 | 1225 | 0.86 | 0.85 | 0.75 | 0.84 |
| | Non | 167 | 2030 | 2197 | | | | |
| | Total | 1082 | 2340 | | | | | |
| $\text{maize}_{10-20}$ | Irrigated | 875 | 321 | 1196 | 0.86 | 0.83 | 0.73 | 0.83 |
| | Non | 175 | 2052 | 2227 | | | | |
| | Total | 1050 | 2373 | | | | | |

**Point 5:** Some typos are found in manuscript, and check manuscript carefully and correct them. e.g. Line143: delete 'in China'.

**Response:** Thank you for your careful comments. We have modified all typos in the revised paper (Line 58, 100, 143, 524).

Line 58: $r^2$ -> $R^2$

Line 100: "accumulated precipitation for 10 days" -> "ante-accumulated precipitation over ten days"

Line 143:

Line 524: "mode" -> "factor"

**Point 6:** Figure 2 should be improved. Currently, some labels are too vague to clearly identify.

**Response:** Many thanks for your advice. We have modified it in the revised paper.

**Point 7:** Please modify the line widths in Table 2.

**Response:** Many thanks for your careful check. We have modified it in the revised paper.

**Point 8:** Line257: insert blank between two words. 'Figure8' -> 'Figure 8'.

**Response:** Thanks for your careful review. We have modified it in the revised paper. (Line 284).

**Point 9:** Figure S5 was not used in the main text, please cite it in main text or delete it from supplemental material.

**Response:** Thanks for your careful review. We have deleted it from supplemental material.

**Dear Editor and Reviewer # 3:**

We appreciate your insightful comments on our paper. The comments offered have been immensely helpful. We have responded to every question, indicating exactly how we addressed each concern or problem and describing the changes we have made. The revisions have been approved by all authors. The point-to-point responses to your comments are listed below in **blue**.

This is an interesting effort in developing the SM product for crop dryland, which has potential for various applications. The paper is well written and organized. Taking the CIR as a predictor seems to be a useful way to predict SM in crop dryland. However, I have some concerns as following. Please pay more attention on the comments about line 174-175.

**Point 1:** Why only mapping SM for dryland, not rice?

**Response:** Rice is commonly grown in southern areas with plenty rainfall or northern areas well equipped by irrigation in China. Therefore, soil moisture is usually over saturated and keeps constant (near 100%) during the whole growing season (Zheng et al., 2000; Alhaj Hamoud et al., 2019). Considering the significant role of SM for maize and wheat planted in dryland across China, we mapped the SM for crop drylands, not including rice.

Reference:
Alhaj Hamoud, Y., Guo, X., Wang, Z., Shaghaleh, H., Chen, S., Hassan, A., and Bakour, A.: Effects of irrigation regime and soil clay content and their interaction on the biological yield, nitrogen uptake and nitrogen-use efficiency of rice grown in southern China, Agricultural Water Management, 213, 934–946, https://doi.org/10.1016/j.agwat.2018.12.017, 2019.

Zheng, X., Wang, M., Wang, Y., Shen, R., Gou, J., Li, J., Jin, J., and Li, L.: Impacts of soil moisture on nitrous oxide emission from croplands: a case study on the rice-based agro-ecosystem in Southeast China, Chemosphere - Global Change Science, 2, 207–224, https://doi.org/10.1016/S1465-9972(99)00056-2, 2000.

**Point 2:** Line 110-115: there are two sources of FC which one is used?

**Response:** Field capacity (fc) was obtained from OpenLandMap which included fc under 33kPa at 0cm (b0) and 10 cm (b10) depth. When predict $ChinaCropSM_{0-10cm}$, we used fc under 33kPa at 0cm (b0) depth. When predict $ChinaCropSM_{10-20cm}$, we used fc under 33kPa at 10cm (b10) depth.

**Point 3:** Line 120: the short name "AMS" is used only one time. Consider full name. In addition, what is R4, R5 and R16? And it should not be calculated only for AMS but for each cell, as a predictor.

**Response:** Many thanks for your advice. We have increased the full name of "AMS" in the revised paper.

Yes, R4, R5 and R16 is calculated for each cell, as a predictor. Actually, the R4, R5 and R16 are river network vector data at different levels in China. When training sample data, we calculate the distance for AMS. Additionally, we calculated the distance from each cell to river network vector data when predicting the ChinaCropSM.

**Point 4:** Line 171: Grammar error. Not a complete sentence.

**Response:** Thank you for your careful comments. We have modified it (Line 180).

"As for the response variable (Classified Irrigation CIR), it is calculated by irrigation threshold (Table 2) and in situ information, including crop type, phenology and soil depth.".

**Point 5:** Line 174-175: It should not be random splitting because SM of different time from the same site may be highly correlated. This will give a higher performance for the model. Instead, the splitting should be based on sites, i.e., data from a site should be all in the training set or all in test set. Note that the model is predicting unknown locations based on the observing sites, and the spatial interpolation ability should be evaluated by the site-based splitting.

**Response:** Thanks very much for your constructive comment.

According to your site-based splitting method, we re-optimized the hyper-parameters of the prediction model to reduce overfitting and evaluated the prediction results. We found the soil moisture predicted by your method agreed well with in situ SM observations (ubRMSE ranges from 0.046–0.057, and $R^2$ ranges from 0.642–0.761), although the model performance drops slightly (Figure 1).

Similarly, in the case of site-based splitting, all prediction accuracy of SM were consistently improved both for crops and depths with comparison of those without an irrigation module (e.g. $R^2$ increased by 9–41%, ubRMSE decreased by 21–26%) (Figure 2). Also, we further compared our ChinaCropSM1km with the two popular public global SM products (Table 1). All indexes of our ChinaCropSM were consistently indicated by the higher accuracy.

Different splitting methods during training and testing do affect model performance. Selecting which splitting method to improve the generalization performance is dependent on data. Generally, the larger size of data, the smaller effect of the splitting methods on the results (Birba, 2020). Therefore, the model performances of two splitting methods show no significant differences because of quantities of field observations available in our study. We have followed you to insert deeper and more extent discussions into our manuscript (Line 307~322 in the revised manuscript).

Reference:

Birba, D. E.: A Comparative study of data splitting algorithms for machine learning model selection, 2020.

The results are following:

[Figure]

Figure 1 Comparison between the predicted soil moisture (ChinaCropSM1km) and in situ samples by crops and depths (cm) according to site-based splitting.

[Figure]

Figure 2 Comparison of soil moisture accuracy between with irrigation and without an irrigation module according to site-based splitting.

Table 1 Summary on means of evaluation indexes of three products (ChinaCropSM1km, RSSSM and ESA CCI SM).

| Product | ChinaCropSM | RSSSM | ESA CCI SM |
|---|---|---|---|
| $r$ | 0.85 | 0.52 | 0.42 |
| RMSE | 0.054 | 0.144 | 0.120 |
| bias | −0.005 | −0.112 | −0.066 |
| ubRMSE | 0.054 | 0.092 | 0.100 |

**Point 6:** Line 185: How many times do you run the model to get the importance, as the importance will be different each time. It should take the average importance of dozens of runs like 100.

**Response:** Yes, we did take the averages of dozens of runs. We ran each training model 50 times to get the importance and averaged the importance outcome.

**Point 7:** Fig.6 and 7: what are the different boxes stand for?

**Response:** The boxes in Fig.6 and Fig.7 actually stand for different results, with spatial pattern in Fig.6 and temporal one in Fig. 7. Both patterns were conducted between ChinaCropSM1km and the in situ SM observations.

The horizontal line within each box stands for median, the white dot for mean, the box bottom for first quantile, the top for third quantile, and black dots for outliers.

**Point 8:** Section 3.5: I do not think this comparison is fare. The evaluation using the test data for Cropland should be used instead of all in situ data because the model used them to establish leading to an independent evaluation.

**Response:** Actually, we only used the testing data for evaluating, not including all in situ data. We agreed well with you that using all observations will lead to an independent evaluation.

**Dear Editor and Reviewer # 3:**

We appreciate your insightful comments on our paper. The comments offered have been immensely helpful. We have responded to every question, indicating exactly how we addressed each concern or problem and describing the changes we have made. The revisions have been approved by all authors. The point-to-point responses to your comments are listed below in **blue**.

This is an interesting effort in developing the SM product for crop dryland, which has potential for various applications. The paper is well written and organized. Taking the CIR as a predictor seems to be a useful way to predict SM in crop dryland. However, I have some concerns as following. Please pay more attention on the comments about line 174-175.

**Point 1:** Why only mapping SM for dryland, not rice?

**Response:** Rice is commonly grown in southern areas with plenty rainfall or northern areas well equipped by irrigation in China. Therefore, soil moisture is usually over saturated and keeps constant (near 100%) during the whole growing season (Zheng et al., 2000; Alhaj Hamoud et al., 2019). Considering the significant role of SM for maize and wheat planted in dryland across China, we mapped the SM for crop drylands, not including rice.

Reference:
Alhaj Hamoud, Y., Guo, X., Wang, Z., Shaghaleh, H., Chen, S., Hassan, A., and Bakour, A.: Effects of irrigation regime and soil clay content and their interaction on the biological yield, nitrogen uptake and nitrogen-use efficiency of rice grown in southern China, Agricultural Water Management, 213, 934–946, https://doi.org/10.1016/j.agwat.2018.12.017, 2019.

Zheng, X., Wang, M., Wang, Y., Shen, R., Gou, J., Li, J., Jin, J., and Li, L.: Impacts of soil moisture on nitrous oxide emission from croplands: a case study on the rice-based agro-ecosystem in Southeast China, Chemosphere - Global Change Science, 2, 207–224, https://doi.org/10.1016/S1465-9972(99)00056-2, 2000.

**Point 2:** Line 110-115: there are two sources of FC which one is used?

**Response:** Field capacity (fc) was obtained from OpenLandMap which included fc under 33kPa at 0cm (b0) and 10 cm (b10) depth. When predict $ChinaCropSM_{0-10cm}$, we used fc under 33kPa at 0cm (b0) depth. When predict $ChinaCropSM_{10-20cm}$, we used fc under 33kPa at 10cm (b10) depth.

**Point 3:** Line 120: the short name "AMS" is used only one time. Consider full name. In addition, what is R4, R5 and R16?  And it should not be calculated only for AMS but for each cell, as a predictor.

**Response:** Many thanks for your advice. We have increased the full name of "AMS" in the revised paper.

Yes, R4, R5 and R16 is calculated for each cell, as a predictor. Actually, the R4, R5 and R16 are river network vector data at different levels in China. When training sample data, we calculate the distance for AMS. Additionally, we calculated the distance from each cell to river network vector data when predicting the ChinaCropSM.

**Point 4:** Line 171: Grammar error. Not a complete sentence.

**Response:** Thank you for your careful comments. We have modified it (Line 180).

"As for the response variable (Classified Irrigation CIR), it is calculated by irrigation threshold (Table 2) and in situ information, including crop type, phenology and soil depth.".

**Point 5:** Line 174-175: It should not be random splitting because SM of different time from the same site may be highly correlated. This will give a higher performance for the model. Instead, the splitting should be based on sites, i.e., data from a site should be all in the training set or all in test set. Note that the model is predicting unknown locations based on the observing sites, and the spatial interpolation ability should be evaluated by the site-based splitting.

**Response:** Thanks very much for your constructive comment.
According to your site-based splitting method, we re-optimized the hyper-parameters of the prediction model to reduce overfitting and evaluated the prediction results. We found the soil moisture predicted by your method agreed well with in situ SM observations (ubRMSE ranges from 0.046–0.057, and $R^2$ ranges from 0.642–0.761), although the model performance drops slightly (Figure 1).

Similarly, in the case of site-based splitting, all prediction accuracy of SM were consistently improved both for crops and depths with comparison of those without an irrigation module (e.g. $R^2$ increased by 9–41%, ubRMSE decreased by 21–26%) (Figure 2). Also, we further compared our ChinaCropSM1km with the two popular public global SM products (Table 1). All indexes of our ChinaCropSM were consistently indicated by the higher accuracy.

Different splitting methods during training and testing do affect model performance. Selecting which splitting method to improve the generalization performance is dependent on data. Generally, the larger size of data, the smaller effect of the splitting methods on the results (Birba, 2020). Therefore, the model performances of two splitting methods show no significant differences because of quantities of field observations available in our study. We have followed you to insert deeper and more extent discussions into our manuscript (Line 307~322 in the revised manuscript).

Reference:

Birba, D. E.: A Comparative study of data splitting algorithms for machine learning model selection, 2020.

The results are following:

[Figure]

Figure 1 Comparison between the predicted soil moisture (ChinaCropSM1km) and in situ samples by crops and depths (cm) according to site-based splitting.

[Figure]

Figure 2 Comparison of soil moisture accuracy between with irrigation and without an irrigation module according to site-based splitting.

Table 1 Summary on means of evaluation indexes of three products (ChinaCropSM1km, RSSSM and ESA CCI SM).

| Product | ChinaCropSM | RSSSM | ESA CCI SM |
|---|---|---|---|
| $r$ | 0.85 | 0.52 | 0.42 |
| RMSE | 0.054 | 0.144 | 0.120 |
| bias | –0.005 | –0.112 | –0.066 |
| ubRMSE | 0.054 | 0.092 | 0.100 |

**Point 6:** Line 185: How many times do you run the model to get the importance, as the importance will be different each time. It should take the average importance of dozens of runs like 100.

**Response:** Yes, we did take the averages of dozens of runs. We ran each training model 50 times to get the importance and averaged the importance outcome.

**Point 7:** Fig.6 and 7: what are the different boxes stand for?

**Response:** The boxes in Fig.6 and Fig.7 actually stand for different results, with spatial pattern in Fig.6 and temporal one in Fig. 7. Both patterns were conducted between ChinaCropSM1km and the in situ SM observations.

The horizontal line within each box stands for median, the white dot for mean, the box bottom for first quantile, the top for third quantile, and black dots for outliers.

**Point 8:** Section 3.5: I do not think this comparison is fare. The evaluation using the test data for Cropland should be used instead of all in situ data because the model used them to establish leading to an independent evaluation.

**Response:** Actually, we only used the testing data for evaluating, not including all in situ data. We agreed well with you that using all observations will lead to an independent evaluation.

**Dear Topical Editor :**

We appreciate your insightful comments on our paper. The comments offered have been immensely helpful. We have responded to every question, indicating exactly how we addressed each concern or problem and describing the changes we have made. The revisions have been approved by all authors. The point-to-point responses to your comments are listed below in **blue**.

**Comments to the author:**

Before the formal acceptance, the authors need clarify several issues:

**Point 1:** The wheat and maize distribution maps were constant, which inevitably brought uncertainties. In fact, there are currently available related products including those developed by the authors. Therefore, at least, the authors need give a paragraph or several statements discussing the limits in the datasets and the methodology.

**Response:** Thanks very much for your constructive comment. We have followed you to insert deeper and more extent discussions into the dataset and the methodology sections (Line 303~308).

"Third, to provide more extensive SM data as possible as we can, a constant layer integrated with all pixels planting wheat/maize during 2000−2015 (http://dx.doi.org/10.17632/jbs44b2hrk.2) was applied to generate our ChinaCropSM1 km. Such merged areas could lead to uncertainties in their spatial distributions because annual wheat/maize planting areas are dynamic over time. To avoid the uncertainties, potential users should mask our products with explicitly annual wheat/maize planting maps to obtain accurate SM data including spatial dynamic information."

**Point 2:** The first comment by Reviewer 2 was not well addressed.

*There is a problem with the resolution. The ground observation data is point measurement data, how to match the resolution of 1km? Please explain this in the*

*manuscript.*

**Response:** Extending pointed-observations into gridded-results (e.g. 1 km) is accepted widely by many studies at a larger region scale (Hengl et al., 2017; He et al., 2022). For example, assessing rice cultivar suitability by crop model simulating at points have successfully applied into gridded areas mainly planted by maize across China (Zhang et al., 2022b); estimating wheat productions by training model at sampling sites has been used to accurately retrieve the gridded-area and gridded-yield of wheat worldwide (Luo et al., 2022); gridded-phenological information has been retrieved successfully by extending pointed-observation into main areas planted by rice in Asia (Zhang et al., 2022a). We are sure, up to date, the extending method is reasonable and robust, and has been recognized as an alternative way for conducting larger scale study. As for the resolution, it is strongly depended on the quality of independent variables (e.g. climate, environment and remote sensing products) and ground observations.

As for the quality of our ground observations, we inserted the relevant text in the revised manuscript (Please check our revisions in Lines 91-95 in bold):

The in situ SM observation data (http://data.cma.gov.cn/data/detail/dataCode/AGME_AB2_CHN_TEN.html, last accessed: 18 April 2021) from 1993 to 2018 were obtained from agricultural meteorological sites (AMS) in China, which recorded the location, crop type, phenology, soil depth and SM. SM was measured at the depths of 10 cm and 20 cm at each AMS on the 8th, 18th and 28th of each month. For each sample, crop phenology was observed and recorded by well-trained agricultural technicians in experimental fields (the average field size was 0.15 ha) and then were checked and qualified by the Chinese Agricultural Meteorological Monitoring System (CAMMS). **"The location of AMS is generally selected in areas with relatively homogeneous soil properties. Also the fact that crops were quite well managed by irrigation according to weather variability and crop growth status makes the crop SM records largely representative the overall level of pixels (1 km×1 km) (Zhang et al., 2020; Li et al., 2021)."** The first layer (0–10 cm) has been widely used to investigate spatial and

temporal characteristics of SM and validate SM retrieved from microwave across China.

As for how to extend our pointed observations into 1 km grids, we followed the below steps:"We used the Extract Values to Points tool to extract the 1 km resolution raster information of the environmental (i.e., SP, RSD and GI) data to AMS point data, output point data attributes and save it in CSV format to obtain a dataset of environmental factors through ArcGIS 10.5.". "All these point samples were used to develop the pointed SM model, and then these pointed models are applied to inversely calculate the gridded SM by inputting 1-km raster environmental variables.".

**Reference:**

He, Q., Wang, M., Liu, K., Li, K., and Jiang, Z.: GPRChinaTemp1km: a high-resolution monthly air temperature data set for China (1951–2020) based on machine learning, Earth Syst. Sci. Data, 14, 3273–3292, https://doi.org/10.5194/essd-14-3273-2022, 2022.

Hengl, T., Mendes de Jesus, J., Heuvelink, G. B. M., Ruiperez Gonzalez, M., Kilibarda, M., Blagotić, A., Shangguan, W., Wright, M. N., Geng, X., Bauer-Marschallinger, B., Guevara, M. A., Vargas, R., MacMillan, R. A., Batjes, N. H., Leenaars, J. G. B., Ribeiro, E., Wheeler, I., Mantel, S., and Kempen, B.: SoilGrids250m: Global gridded soil information based on machine learning, PLoS ONE, 12, e0169748, https://doi.org/10.1371/journal.pone.0169748, 2017.

Li, Z., Zhang, Z., and Zhang, L.: Improving regional wheat drought risk assessment for insurance application by integrating scenario-driven crop model, machine learning, and satellite data, Agricultural Systems, 191, 103141, https://doi.org/10.1016/j.agsy.2021.103141, 2021.

Luo, Y., Zhang, Z., Cao, J., Zhang, L., Zhang, J., Han, J., Zhuang, H., Cheng, F., and Tao, F.: Accurately mapping global wheat production system using deep learning algorithms, International Journal of Applied Earth Observation and Geoinformation, 110, 102823, https://doi.org/10.1016/j.jag.2022.102823, 2022.

Zhang, J., Wu, H., Zhang, Z., Luo, Y., Han, J., and Tao, F.: Asian Rice Calendar Dynamics Detected by Remote Sensing and Their Climate Drivers, Remote Sensing,

14, 4189, https://doi.org/10.3390/rs14174189, 2022a.

Zhang, L., Zhang, Z., Tao, F., Luo, Y., and Cao, J.: Adapting to climate change precisely through cultivars renewal for rice production across China: When, where, and what cultivars will be required?, Agricultural and Forest Meteorology, 316, 108856, https://doi.org/10.1016/j.agrformet.2022.108856, 2022b.

Zhang, Z., Li, Z., Chen, Y., Zhang, L., and Tao, F.: Improving regional wheat yields estimations by multi-step-assimilating of a crop model with multi-source data, Agricultural and Forest Meteorology, 290, 107993, https://doi.org/10.1016/j.agrformet.2020.107993, 2020.

**Point 3:** The final product only covers wheat and maize while the title writes all crop drylands. The authors need either revise the title or really complete retrievals for all types of crop drylands.

**Response:** Thanks for your suggestion. We have revised the title to "**ChinaCropSM1 km: a fine 1 km daily Soil Moisture dataset for dryland wheat and maize across China during 1993–2018**".

We have made corresponding modifications in the revised paper, and we have declared that crop refers specifically to maize and wheat, please see in Figure 2 and Data availability.

[Figure]

Figure 2 Flow chart for producing ChinaCropSM1 km.

Data availability:

The 1 km gridded daily soil moisture dataset for main crops (i.e., wheat and maize) dryland in China from 1993 to 2018 (ChinaCropSM1 km) are publicly available at https://zenodo.org/record/6834530 (wheat$_{0–10}$) (Cheng et al., 2022a), https://zenodo.org/record/6822591 (wheat$_{10–20}$) (Cheng et al., 2022b), https://zenodo.org/record/6822581 (maize$_{0–10}$) (Cheng et al., 2022c) and https://zenodo.org/record/6820166 (mazie$_{10–20}$) (Cheng et al., 2022d).

**Point 4:** There are still many language errors or typos in the text (e.g., the usage of "however" in Line 55 and "still further" in Line 239).

**Response:** Thank you a lot for the suggestion. We have revised the text with the assistance from a native English speaker who is a competent technical writer. The language has been improved throughout the manuscript. Also, we provided the certificate for the language editing below. Please see the revised manuscript for more details.

[Figure]

**Editing Certificate**

This document certifies that the manuscript

**ChinaCropSM1 km: a fine 1 km daily Soil Moisture dataset for dryland wheat and maize across China during 1993–2018**

prepared by the authors

**Fei Cheng, Zhao Zhang, Huimin Zhuang, Jichong Han, Yuchuan Luo, Juan Cao, Liangliang Zhang, Jing Zhang, Jialu Xu and Fulu Tao**

was edited for proper English language, grammar, punctuation, spelling, and overall style by one or more of the highly qualified native English speaking editors at AJE.

This certificate was issued on **December 5, 2022** and may be verified on the AJE website using the verification code **C182-CE5F-0C5A-5FCD-2732**.

[Figure]

Neither the research content nor the authors' intentions were altered in any way during the editing process. Documents receiving this certification should be English-ready for publication; however, the author has the ability to accept or reject our suggestions and changes. To verify the final AJE edited version, please visit our verification page at aje.com/certificate. If you have any questions or concerns about this edited document, please contact AJE at support@aje.com.

AJE provides a range of editing, translation, and manuscript services for researchers and publishers around the world. For more information about our company, services, and partner discounts, please visit aje.com.

Figure R1. Certificate of editing